# 21st century ocean forcing of the Greenland Ice Sheet for modeling of sea level contribution

Donald A. Slater[1], Denis Felikson[2], Fiamma Straneo[1], Heiko Goelzer[3,4], Christopher M. Little[5], Mathieu Morlighem[6], Xavier Fettweis[7], and Sophie Nowicki[2]

[1]Scripps Institution of Oceanography, University of California San Diego, La Jolla, CA, USA
[2]Cryospheric Sciences Laboratory, NASA Goddard Space Flight Center, Greenbelt, MD, USA
[3]Institute for Marine and Atmospheric research Utrecht, Utrecht University, Utrecht, the Netherlands
[4]Laboratoire de Glaciologie, Université Libre de Bruxelles, Brussels, Belgium
[5]Atmospheric and Environmental Research, Inc., Lexington, MA, USA
[6]Department of Earth System Science, University of California, Irvine, USA
[7]Laboratory of Climatology, Department of Geography, University of Liège, Liège, Belgium

**Correspondence:** Donald Slater (daslater@ucsd.edu)

**Abstract.** Changes in ocean temperature and salinity are expected to be an important determinant of the Greenland Ice Sheet's future sea level contribution. Yet simulating the impact of these changes in continental-scale ice sheet models remains challenging due to the small scale of key physics, such as fjord circulation and plume dynamics, and poor understanding of critical processes, such as calving and submarine melting. Here we present the ocean forcing strategy for Greenland Ice Sheet models
taking part in the Ice Sheet Model Intercomparison Project for CMIP6 (ISMIP6), the primary community effort to provide 21st century sea level projections for the Intergovernmental Panel on Climate Change 6th Assessment Report. Beginning from global atmosphere-ocean general circulation models, we describe two complementary approaches to provide ocean boundary conditions for Greenland Ice Sheet models, termed the 'retreat' and 'submarine melt' implementations. The retreat implementation parameterises glacier retreat as a function of projected subglacial discharge and ocean thermal forcing, is designed to
be implementable by all ice sheet models and results in retreat of around 1 and 15 km by 2100 in RCP2.6 and 8.5 scenarios, respectively. The submarine melt implementation provides estimated submarine melting only, leaving the ice sheet model to solve for the resulting calving and glacier retreat and suggests submarine melt rates will change little under RCP2.6 but will approximately triple by 2100 under RCP8.5. Both implementations have necessarily made use of simplifying assumptions and poorly-constrained parameterisations and as such, further research on submarine melting, calving and fjord-shelf exchange
should remain a priority. Nevertheless, the presented framework will allow an ensemble of Greenland Ice Sheet models to be systematically and consistently forced by the ocean for the first time and should result in a significant improvement in projections of the Greenland Ice Sheet's contribution to future sea level change.

## 1 Introduction

The rapid response of the Greenland Ice Sheet to climate warming in the past few decades, together with expectations of
future climate change, have raised concern that Greenland will contribute significantly to sea level change over the coming

decades and centuries (Shepherd et al., 2012; Church et al., 2013; Nick et al., 2013). Greenland contributed ~13.7 mm to global mean sea level between 1972 and 2018, with surface mass balance comprising 35-60% of this ice mass loss (van den Broeke et al., 2016; Mouginot et al., 2019). The remainder derives from discharge from tidewater outlet glaciers, most of which have retreated, accelerated and thinned in recent decades (Rignot and Kanagaratnam, 2006; Khan et al., 2014; Murray et al., 2015). These tidewater glaciers are understood to have responded to climate forcing occurring at their calving fronts, where the ice sheet meets the ocean (Nick et al., 2009; Luckman et al., 2015; Wood et al., 2018). Thus, processes at the ice-ocean boundary and their representations in ice sheet models are a critical component of accurate future sea level projections.

The Ice Sheet Model Intercomparison Project (ISMIP6; Nowicki et al., 2016) is the community-leading effort projecting future sea level contribution from the Greenland and Antarctic Ice Sheets for the coming 6th Assessment Report of the Intergovernmental Panel on Climate Change (IPCC AR6). ISMIP6 follows a history of similar initiatives, such as SeaRISE (Bindschadler et al., 2013; Nowicki et al., 2013a, b) and ice2sea (Gillet-Chaulet et al., 2012; Goelzer et al., 2013), aimed at bringing together a number of ice sheet models and scientists across disciplines to improve projections of ice sheet mass loss. Compared to previous initiatives, ISMIP6 is the first such effort to be fully integrated within the Coupled Model Intercomparison Project (CMIP6); CMIP6 being itself a model intercomparison exercise focused on the representation of climate in coupled Atmosphere and Ocean general circulation models (AOGCMs). Full details and the experimental protocol of the ISMIP6 project can be found in Nowicki et al. (2016) and Nowicki et al. (2020), while the Greenland Ice Sheet sea level projections can now be found in Goelzer et al. (2020). The present paper focuses specifically on one aspect of ISMIP6: the representation of ocean forcing in the simulations of the Greenland Ice Sheet. The aim is to relate large-scale climate – as defined by the CMIP AOGCMs – to an ocean boundary condition for the ice sheet models.

Ocean forcing of the Greenland Ice Sheet occurs at around 300 approximately vertical glacier calving fronts around Greenland and at several larger ice shelves and floating ice tongues located in the far north. Ocean forcing is here broadly defined as melting of the ice-ocean boundary (hereafter called submarine melting) and the impact of this melting on calving and glacier retreat. The design of boundary conditions that represent ocean forcing must take into account three sets of processes. First, the transport of ocean heat from the far-field ocean to calving fronts across continental shelves and up long and narrow fjords. Second, the near-ice circulation which drives heat transfer through the ice-ocean boundary. Third, the impact of submarine melting on iceberg calving and glacier retreat.

Understanding of these key processes has advanced through both observations and models. Considering observations, warm Atlantic-origin water is found on the continental shelf around Greenland either due to transport from the deep ocean to the shelf, often in deep troughs, or due to advection along the shelf by coastal currents (Sutherland et al., 2013; Rykova et al., 2015; Schaffer et al., 2017). The same waters are found adjacent to calving fronts (Straneo et al., 2012) and may enter the fjords by numerous processes including a glacier-driven estuarine-type circulation that may be prevalent in summer (Motyka et al., 2003; Gladish et al., 2015), fjord-shelf exchange driven by winds both inside and outside of fjords (Jackson et al., 2014; Spall et al., 2017) and exchange due to variability in shelf water properties (Mortensen et al., 2014; Carroll et al., 2018). Once warm water reaches the calving front, the transfer of heat across the ice-ocean boundary layer is promoted by the near-ice circulation (Holland and Jenkins, 1999). During summer, the release of ice sheet meltwater into fjords from beneath tidewater

glaciers drives localised but vigorous upwelling plumes which are thought to drive rapid submarine melting (Mankoff et al., 2016; Sutherland et al., 2019). These plumes may also fuel a fjord-wide circulation which enhances submarine melting over the full calving front (Slater et al., 2018; Kienholz et al., 2019). Submarine melting can shape the calving front, creating regions of undercut and overcut ice (Fried et al., 2019), which may in turn enhance iceberg calving and drive glacier retreat (Luckman

et al., 2015; How et al., 2019). Greenland's shelf and fjords, however, remain sparsely observed, especially in winter, and we have very few observations of submarine melt rate. Similarly, significant uncertainty surrounds the dynamic impact of submarine melting on calving due to the difficulty of making the necessary measurements close to dangerous calving fronts.

       Many of these processes can be captured by models at the individual fjord or glacier scale. Cowton et al. (2016) and Fraser et al. (2018) have modeled fjord-shelf exchange at Kangerdlugssuaq Fjord in south-east Greenland, while Carroll et al. (2017)

modeled fjord water renewal driven by subglacial discharge in an idealised domain. Plumes and the near-ice circulation they generate have been captured by models focused on the part of the fjord within a few kilometers of the calving front (Xu et al., 2012; Slater et al., 2018). The impact of submarine melting on calving has been studied at high resolution in both idealised and realistic settings (Cowton et al., 2019; Ma and Bassis, 2019; Todd et al., 2019). Yet, the model resolution in these studies, at ∼500 m for the fjord simulations, ∼10 m for the plume simulations and ∼50 m for the calving simulations, is far smaller

than the ∼50 km resolution of AOGCMs (e.g. Watanabe et al., 2010) or ∼2 km resolution of Greenland Ice Sheet models (Goelzer et al., 2018). Even regional ocean models (e.g. Gillard et al., 2016) do not yet represent fjords and fjord processes. Thus, climate and ice sheet models do not have sufficient resolution to capture the processes that modulate the effect of the ocean on the Greenland Ice Sheet.

       At present, therefore, projecting the sea level contribution of the Greenland Ice Sheet requires that we parameterise ice-

ocean processes, but well-validated parameterisations are not readily available. While progress has been made in observing and modeling fjord circulation and fjord-shelf exchange, we as yet lack simple parameterisations or box models that could represent these processes in an efficient fashion (i.e. without resorting to computationally expensive hydrodynamic models). Conversely, parameterisations exist for the submarine melting induced by plumes (Rignot et al., 2016; Slater et al., 2016) but we still have few observations with which to validate these parameterisations (Sutherland et al., 2019). Lastly, the search for a

universal calving law has a long history (e.g. Benn et al., 2007) but as for submarine melting no calving parameterisation has undergone sufficient validation for confident use.

       Given the described process uncertainty, the small scale of key processes and the current lack of parameterisations for these processes, projecting ocean-induced ice mass loss from the Greenland Ice Sheet is very challenging. To date, attempts to project future ice discharge from tidewater glaciers have often relied on extrapolation from a few glaciers to the whole ice sheet

(Goelzer et al., 2013; Nick et al., 2013; Peano et al., 2017; Beckmann et al., 2019; Morlighem et al., 2019) or have employed ad-hoc methods to mimic the impact of ocean forcing that are not easily relatable to climate warming scenarios (Price et al., 2011; Bindschadler et al., 2013; Fürst et al., 2015). In a single ice sheet model, a significant advance was recently made by Aschwanden et al. (2019), who ran full ice sheet projections that resolve tidewater glaciers and were forced by estimated submarine melt rates but many of the ice sheet models taking part in ISMIP6 do not currently have the resolution or technical

capability for this approach (Goelzer et al., 2018).

In spite of the described difficulties, we present a strategy for simulating the impact of the ocean on the ice sheet that will enable a suite of Greenland Ice Sheet models of diverse capabilities to be systematically forced by future warming scenarios (Goelzer et al., 2020). We do not aim to solve the problems of process understanding, scale and parameterisation but rather to offer a pragmatic approach based on the current state of knowledge. This approach draws on existing parameterisations for

tidewater glacier retreat (Slater et al., 2019) and submarine melting (Rignot et al., 2016). The paper proceeds as follows. An overview of the two-tier strategy for ocean forcing is given, before the subglacial runoff and ocean thermal forcing datasets are described. These time series are combined into projections of glacier retreat and submarine melting. We finally discuss the projected ocean forcing, its temporal evolution and spatial and inter-model variability.

## 2  Methods

### 2.1  Overview

We develop two possible implementations for ocean forcing of Greenland Ice Sheet models, referred to as the retreat implementation and the submarine melt implementation (Fig. 1). The retreat implementation is designed to be implementable by all of the ice sheet models taking part in ISMIP6 regardless of resolution, model physics or spin-up procedure. In this implementation, retreat of the ice-ocean boundary is estimated as a linear function of parameterised submarine melting (Slater et al.,

2019) and is imposed on an ice sheet model through a time-variable ice mask, an approach first suggested by Cowton et al. (2018). The submarine melt implementation instead provides ice sheet modeling groups with fields of subglacial runoff and ocean properties together with a suggested parameterisation for estimating submarine melt from these quantities. Since glacier retreat is given by a competition between frontal ice velocity, calving and submarine melting, the retreat implementation heavily parameterises ocean forcing by implicitly assuming that all quantities are proportional to submarine melt rate (Slater et al.,

2019). The submarine melt implementation allows ice sheet models to resolve the competition between velocity, calving and melting, perhaps by implementing a calving law that depends on submarine melt rate.

Both implementations require a parameterisation for submarine melting. Theoretical considerations suggest that melt rates are controlled primarily by local ocean velocity and ocean thermal forcing, the latter defined as the difference between the in-situ temperature and in-situ freezing point (Gade, 1979; Holland and Jenkins, 1999). Near-ice ocean velocities are thought

to be highest inside vigorous plumes resulting from the emergence of buoyant subglacial runoff from the grounding line of the glacier (Mankoff et al., 2016). Submarine melt rate parameterisations (Jenkins, 2011; Xu et al., 2013; Slater et al., 2016), therefore, typically include the basic ingredients of subglacial runoff ($Q$) and ocean thermal forcing (TF). In the retreat implementation we follow Slater et al. (2019) in assuming that submarine melting is proportional to $Q^{0.4}$TF and retreat ($\Delta L$, in km) is proportional to submarine melting, so that retreat may be estimated as

$$\Delta L = \kappa \Delta (Q^{0.4}\,\mathrm{TF}) \tag{1}$$

where $Q$ is the mean summer (June-July-August) subglacial runoff (in $\mathrm{m}^3\mathrm{s}^{-1}$) and TF is the ocean thermal forcing (in °C). Slater et al. (2019) calibrated the linear coefficient $\kappa$ at nearly 200 tidewater glaciers by considering observed retreat, estimated

subglacial runoff and observed ocean thermal forcing over the time period 1960-2018. This resulted in a distribution for $\kappa$ (in units $\mathrm{km\,(m^3 s^{-1})^{-0.4}\,{}^\circ C^{-1}}$) having a median $\kappa_{50} = -0.17$ and quartiles $\kappa_{25} = -0.37$ and $\kappa_{75} = -0.06$, respectively.

For the submarine melt implementation, we follow Rignot et al. (2016) in parameterizing submarine melt rate ($\dot{m}$) as

$$\dot{m} = (3 \cdot 10^{-4} \, h \, q^{0.39} + 0.15) \, \mathrm{TF}^{1.18} \tag{2}$$

5   where $h$ is grounding line depth (in m), TF is the ocean thermal forcing (in $^\circ$C) and $q$ is the annual mean subglacial runoff normalised by calving front area (in $\mathrm{m\,d^{-1}}$). We acknowledge the inconsistency of using summer runoff for the retreat implementation and annual runoff for the submarine melt implementation but we emphasise that this makes no practical difference since annual and summer runoff are very closely related, even in the future projections when the melt season becomes longer (Slater et al., 2019). The parameterisation for submarine melting is slightly more complex than that for retreat but is functionally 10   very similar.

The chosen parameterisations require the two basic inputs of future subglacial runoff and ocean thermal forcing, which are estimated from CMIP AOGCMs. While it is hoped that some of the new generation of climate models (CMIP6) will be used in ISMIP6, very few CMIP6 simulations were available at the time of writing and given the time constraints of the ISMIP6 project it was decided to focus largely on CMIP5, for which the full ensemble is already available. We consider 6 CMIP5 15   AOGCMs (Table 1) that represent a subset of the full CMIP5 ensemble but emphasise that the process would be identical for CMIP6 inputs. The 6 CMIP5 AOGCMs have been chosen by selecting AOGCMs with minimal biases in the present day and with the aim of sampling the diversity of projected climate change, as described in Barthel et al. (2019). The focus is on the RCP8.5 scenario, a high greenhouse gas emissions pathway in which radiative forcing reaches $8.5\,\mathrm{W\,m^{-2}}$ in 2100 (Riahi et al., 2011; Nowicki et al., 2016). We also consider a single RCP2.6 simulation (radiative forcing of $2.6\,\mathrm{W\,m^{-2}}$ in 2100). Each 20   of the CMIP5 AOGCM simulations covers the period 1850-2100, with 1850-2005 considered the historical spin-up period and the emissions forcing applied from 2006-2100 (Taylor et al., 2012). Ice sheet model ocean forcing is delivered for the time period from 1950-2100. The remainder of this methods section describes the calculation of subglacial runoff and ocean thermal forcing from AOGCM output and the combination of these datasets into ice sheet model ocean forcing in the retreat and submarine melt implementations (Fig. 1).

25  **2.2  Atmosphere**

### 2.2.1   Estimating ice sheet surface runoff using MAR

Since the CMIP5 AOGCMs have a crude representation of ice sheet surface mass balance, the Modèle Atmosphérique Régional (MAR) is used to estimate surface runoff by downscaling the CMIP5 AOGCM atmospheric fields (Fig. 1; Fettweis et al. (2013)). The most recent version of the model, MAR 3.9.6, is run at 15 km resolution with surface mass balance components 30   (including runoff) statistically downscaled afterwards to 1 km (Franco et al., 2012) to better account for sub-grid topography (Fig. 2a). Each simulation is forced at its boundaries by 6-hourly output from a CMIP5 AOGCM (Table 1) over the period 1950-2100.

### 2.2.2 Hydrological drainage basins

Both the retreat and submarine melt implementations use an estimate of subglacial runoff per tidewater glacier, which requires a hydrological drainage basin for each glacier (Fig. 1). These basins are delineated based on the hydrological potential (Shreve, 1972):

$$\phi = \rho_w gb + f\rho_i gh \tag{3}$$

where $\rho_w = 1000\ \mathrm{kg\,m^3}$ and $\rho_i = 910\ \mathrm{kg\,m^3}$ are the densities of freshwater and ice respectively and $g = 9.81\ \mathrm{m^2/s}$ is the gravitational acceleration. Bed topography, $b$ (m) and ice thickness, $h$ (m), come from BedMachinev3 (Morlighem et al., 2017). The variable $f$ represents the ratio of subglacial water pressure to ice overburden pressure. Based on limited borehole pressure records we set $f = 1$ (Meierbachtol et al., 2013; Andrews et al., 2014; Doyle et al., 2018) but acknowledge that different values

of $f$ can alter drainage pathways (e.g. Chu et al., 2016; Moyer et al., 2019). By performing flow routing on $\phi$ (Schwanghart and Scherler, 2014), we identify the area of the ice sheet that drains subglacial water to a given tidewater glacier calving front, defining hydrological drainage basins for each tidewater glacier around the ice sheet (Fig. 2b). For simplicity the hydrological drainage basins are assumed to be constant in time. Given the high density of moulins observed around the margins of the ice sheet during summer (e.g. Yang and Smith, 2016), we assume that all surface meltwater drains to the ice sheet bed close to

where it melts. The subglacial runoff for each glacier can then be estimated by summing the surface runoff from MAR over the hydrological drainage basin for each glacier (Fig. 2b). Studies that have assessed subglacial runoff from fjord observations find agreement between their oceanographic estimates and the method described here (Jackson and Straneo, 2016; Mankoff et al., 2016; Jackson et al., 2017).

### 2.2.3 Present-day bias correction

Many CMIP5 AOGCMs deviate considerably from the observed present-day climate in both the atmosphere and ocean. For example, Menary et al. (2015) show CMIP5 ocean temperature biases can exceed 2°C in the Labrador Sea. If the AOGCM-simulated atmosphere is substantially colder than observations, runoff will be underestimated in MAR when forced by the AOGCM in question (Fettweis et al., 2013). Since in the ISMIP6 exercise we wish to sample uncertainty in future projections rather than the representation of the present day, we perform a bias correction of the projected subglacial runoff at each glacier

to ensure it agrees with our best estimate of present-day runoff (Fig. 1). This bias correction furthermore ensures a continuous transition from present to future forcing, which is desirable as the ice sheet models have been initialised to the present-day forcing (Goelzer et al., 2018).

Present day is defined as the time period 1995-2014. For our best estimate of runoff in the present day we use a 5.5 km resolution regional climate simulation using RACMO2.3p2, forced at its boundaries by ERA-Interim atmospheric reanalysis

(Noël et al., 2018). We ensure that the projected runoff ($Q^{PROJ}$) agrees with the RACMO runoff ($Q^{RACMO}$) in the present day by bias-correcting the projected runoff for each glacier ($j$) as follows:

$$Q_j^{PROJ}(t) \rightarrow Q_j^{PROJ}(t) + \left[ Q_j^{RACMO}(\text{1995-2014}) - Q_j^{PROJ}(\text{1995-2014}) \right] \tag{4}$$

where the 1995-2014 in parentheses indicates the mean value between 1995 and 2014. We assume that the bias remains constant in time. An example of this procedure for Helheim Glacier in SE Greenland under MIROC5 in an RCP8.5 scenario is shown in Fig. 2c. In this case the JJA runoff estimated from MAR forced by MIROC5 is decreased by 55 m$^3$s$^{-1}$ to 316 m$^3$s$^{-1}$ to bring it into agreement with the temporally averaged RACMO2.3p2 output over the period 1995-2014. Note that we do not expect the interannual runoff variability in MAR forced by MIROC5 to agree with RACMO2.3p2 forced by ERA-Interim (Fig. 2, inset) because MIROC5 is a free-running climate model whereas ERA-Interim is an atmospheric reanalysis.

Over all glaciers and all CMIP5 AOGCMs considered (Table 1), the mean bias correction is +2 m$^3$s$^{-1}$ with a standard deviation of 56 m$^3$s$^{-1}$ and a minimum and maximum correction of -527 and +519 m$^3$s$^{-1}$, respectively (Fig. S1). As a fraction of the present-day runoff the mean bias correction is +0.13 with a standard deviation of 0.47. Bias corrections for the largest glacier by ice flux in each sector and for all models are shown in Fig. S1.

It would be better to use MAR forced by ERA-Interim for our best estimate of present day, because it is MAR that is used for the forward projections. If we define the interannual runoff variability as the standard deviation of the detrended projections, we find a mean interannual variability across all glaciers and AOGCMs of 74 m$^3$s$^{-1}$. Given that the bias correction (i.e. the difference between RACMO and MAR in the present day) is typically smaller than the interannual variability of the projections, the use of RACMO for the present day does not cause any inconsistency in practice.

## 2.3  Ocean

### 2.3.1  Defining ocean thermal forcing

Due to a lack of parameterisations that can capture fjord-shelf exchange and fjord circulation without resorting to full hydro-dynamic models, we take a simplified approach to estimating ocean thermal forcing in which the forcing experienced by the glacier is directly related to far-field ocean properties. As such, we are hard-wiring tidewater glaciers to respond to large-scale ocean changes at the expense of most of the local details that we cannot currently account for. Specifically, we spatially aver-age ocean properties over predefined ocean regions and use these properties to force all tidewater glaciers in the same region (Fig. 1). For the retreat implementation, the far-field ocean properties are furthermore depth-averaged (section 2.4.1) while for the submarine melt implementation, the far-field ocean properties are extrapolated into fjords taking account of bathymetry (section 2.5.1).

### 2.3.2  Choice of ice-ocean sectors and spatial averaging

The ice sheet and surrounding ocean were divided into 7 ice-ocean sectors (Fig. 3a) over which ocean properties were spatially averaged (Fig. 1). Each sector is hereafter referred to by its acronym (Fig. 3a), where SW is south-west Greenland, CW is central-west Greenland, NW is north-west Greenland, NO is northern Greenland and similarly for the eastern side of the ice sheet. The sectors, identical to those considered in Slater et al. (2019), were chosen as regions with similar ocean properties largely defined by ocean bathymetry (e.g. Denmark, Fram, Nares Straits) and consistent with the boundaries of commonly used ice sheet drainage basins (e.g. Mouginot et al., 2019) once extended into the ice sheet (see Slater et al. (2019) for a

more in depth description). The small region in CE Greenland is a transition zone between the warm Atlantic waters in the Irminger basin to the south and cool Arctic waters in the Nordic Seas to the north and, as such, was split from the SE and NE Greenland sectors. Each ice-ocean sector extends to the centre of the offshore ocean basin or strait, except for in the Arctic Ocean, Greenland Sea and Labrador Sea where the ocean basin is very large and the sector boundary is located approximately 150 km beyond the shelf break (Fig. 3). With these choices we sample the water masses that interact with the ice sheet but not those that are recirculating (e.g. in western Baffin Bay). Extending the sectors beyond the shelf break also allows us to access many more ocean observations (Fig. S2), which provides greater confidence in the calibration of the retreat parameterisation (Slater et al., 2019) and the bias correction (section 2.3.3). Furthermore, the CMIP5 AOGCM ocean components have coarse resolution of 20 to 100 km around Greenland (e.g. Fig. 3a) and so may not resolve the details of ocean basin to shelf exchange and may have only a few model points on the continental shelf. By extending the sectors beyond the shelf we are allowing the ice sheet ocean forcing to respond to larger-scale ocean features which may be better resolved by the CMIP5 AOGCMs.

To obtain sector ocean properties, monthly CMIP5 AOGCM outputs of modeled ocean potential temperature ($T$) and practical salinity ($S$) are first temporally averaged to annual means (Fig. 3a). Temperature and salinity are then linearly interpolated onto a regular grid with 50 km spatial and 50 m depth resolution (Fig. 3b). Sector ocean properties are finally obtained by taking a simple spatial average over all regular grid points inside a given sector to give a single temperature and salinity profile for each ice-ocean sector for each year (e.g. Fig. 3c).

### 2.3.3 Present-day bias correction

As for the subglacial runoff, we bias-correct the ocean properties to ensure consistency with observations in the present day (Fig. 1). Observations of ocean properties are taken from the Hadley Centre EN4.2.1 dataset (Good et al., 2013), hereafter called EN4. EN4 is a compilation of oceanographic profile data, interpolated onto a monthly 1900-present gridded product available at 1-degree resolution. The coverage of the ocean surrounding Greenland by oceanographic profiles in EN4 during the 1995-2014 present day period is shown in Fig. S2, and indicates that the SE, SW, CE and NE Greenland sectors are relatively well observed while the CW, NW and NO Greenland sectors are sparsely sampled. As such, there is some uncertainty in present-day ocean properties which can feed through to uncertainty in retreat and submarine melt projections (section 4.3).

We obtain annual profiles per ice-ocean sector from EN4 in the same fashion as for the CMIP5 AOGCM projected profiles. While for subglacial runoff we bias-corrected a single value, here we must bias-correct a whole temperature or salinity profile. Rather than applying a different bias correction at each depth level, we apply a single bias correction to the whole profile based on the observed bias in the 200-500 m depth range. Specifically, we bias-correct ocean temperature (Fig. 3c) as follows

$$T_i^{PROJ}(z,t) \rightarrow T_i^{PROJ}(z,t) + \left[ T_i^{EN4}(\text{200-500 m},\text{1995-2014}) - T_i^{PROJ}(\text{200-500 m},\text{1995-2014}) \right] \quad (5)$$

Here, $T_i^{PROJ}(z,t)$ is the projected ocean temperature from the CMIP5 AOGCM in ice-ocean sector $i$ at depth $z$ and in the year $t$. $T_i^{EN4}(\text{200-500 m},\text{1995-2014})$ is the observed ocean temperature in EN4 in sector $i$, depth-averaged between 200 and 500 m and temporally averaged over the 1995-2014 present-day period. $T_i^{PROJ}(\text{200-500 m},\text{1995-2014})$ is the projected ocean temperature from the CMIP5 AOGCM in sector $i$, averaged between 200 and 500 m depth and over the present-day period.

Salinity is bias-corrected in exactly the same fashion. Since the vertical structure of the ocean can vary in time in the CMIP5 AOGCMs, we felt a depth-varying bias correction could lead to unphysical profiles and that a single-valued correction, centered over the depth range most relevant to tidewater glacier grounding lines (Morlighem et al., 2017), was preferable. As for the runoff, the bias correction is assumed constant in time. The magnitude of these corrections can be significant. For example, in MIROC5 RCP8.5 the temperature bias correction for SE Greenland is 1.4°C (Fig. 3c). Over all sectors and CMIP5 AOGCMs considered, the mean temperature bias correction is +0.1°C with a standard deviation of 1.5°C and a minimum and maximum correction of -3.1 and +3.2°C, respectively (Fig. S3).

## 2.4 Retreat implementation

### 2.4.1 Calculation of ocean thermal forcing

To calculate the thermal forcing that enters the retreat parameterisation in Eq. (1), profiles of ocean temperature and salinity (e.g. Fig. 3c) are first converted to profiles of thermal forcing (Fig. 1). The thermal forcing (TF) is for the retreat parameterisation defined as the elevation of the potential ocean temperature $T$ above its local freezing point $T_f$

$$\mathrm{TF}_i(z,t) = T_i(z,t) - T_{f,i}(z,t) = T_i(z,t) - [\lambda_1 S_i(z,t) + \lambda_2 + \lambda_3 z] \tag{6}$$

where in the second equality we have employed a linearised expression for the local freezing point in terms of the practical salinity $S$ and depth $z$ and the constants take values $\lambda_1 = -5.73 \times 10^{-2} \, °\mathrm{C}\,\mathrm{psu}^{-1}$, $\lambda_2 = 8.32 \times 10^{-2} \, °\mathrm{C}$ and $\lambda_3 = 7.61 \times 10^{-4}$ $°\mathrm{C}\,\mathrm{m}^{-1}$ (Jenkins, 2011). As before, $i$ indexes the ice-ocean sector.

In keeping with the simple philosophy of the retreat parameterisation, the profiles of thermal forcing $\mathrm{TF}_i(z,t)$ are finally depth-averaged between 200 and 500 m depth, this being the depth range most relevant to tidewater glacier grounding lines in Greenland (Morlighem et al., 2017). The final thermal forcing entering Eq. (1) in the retreat implementation is a single value per ice-ocean sector per year, for each CMIP5 model considered (Table 1).

### 2.4.2 Glacier-by-glacier projection of retreat

For each CMIP5 AOGCM we first estimate retreat for each of the 191 individual tidewater glaciers considered in Slater et al. (2019) by employing Eq. (1) with the summer subglacial runoff $Q$ per glacier (section 2.2) and ocean thermal forcing TF per sector (section 2.4.1). Specifically, for each glacier $j$ from 1 to 191 we form the time series $Q_j^{0.4}\,\mathrm{TF}_{i(j)}$ where $i(j)$ is the ice-ocean sector $i$ from 1 to 7 in which the glacier $j$ is situated (Fig. 4a). Since this time series has high interannual variability and since for ISMIP6 we are most interested in the multi-decadal sea level contribution, the time series is smoothed using a 20-year centered moving average (Fig. 4a). Lastly, in the CMIP6 and ISMIP6 frameworks (Nowicki et al., 2016; Eyring et al., 2016) the projections begin in 2015 and we project retreat relative to 2014. Thus for each glacier $j$, projected retreat $\Delta L_j(t)$ is given by

$$\Delta L_j(t) = \kappa \left[ Q_j^{0.4}\,\mathrm{TF}_{i(j)}(t) - Q_j^{0.4}\,\mathrm{TF}_{i(j)}(t=2014) \right] \tag{7}$$

where both terms on the right-hand side refer to the smoothed time series. We generate $10^4$ possible future retreat trajectories for each glacier (Fig. 4b) by sampling $10^4$ values of $\kappa$ from its distribution obtained from observations (Slater et al., 2019).

### 2.4.3 Averaging retreat per ice-ocean sector

Due to limitations of the retreat parameterisation - principally its lack of ability to capture individual glacier effects related to bed topography - it is most appropriate to apply retreat averaged over a population of glaciers rather than on an individual glacier basis (Slater et al., 2019). From the ice sheet model perspective, this is also preferable because the state of the ice sheet may differ significantly from the observed ice sheet (Goelzer et al., 2018). Thus, identifying individual glaciers in a given ice sheet model is not trivial so that applying retreat to individual glaciers is also difficult. An obvious solution is to impose a given retreat over a predefined geographical region (or ice-ocean sector), which means averaging retreat over a population of glaciers.

A potential issue is that under the retreat parameterisation (Eq. (1)), glaciers with large hydrological catchments (typically glaciers such as Jakobshavn Isbrae or Helheim) undergo large changes in subglacial runoff and have large projected retreat relative to smaller glaciers. This is considered an important feature of the retreat parameterisation (Slater et al., 2019). Each ice-ocean sector (Fig. 3a) typically has a small number of large glaciers and a large number of small glaciers, such that taking a simple mean of the projected retreat over the glaciers in a sector will result in a trajectory that is much closer to that of the small glaciers than the large glaciers. This is problematic because the primary objective of ISMIP6 is sea level contribution and for Greenland this is dominated by the largest glaciers (Enderlin et al., 2014). To address this problem, we take an ice flux-weighted mean over glaciers in a sector (Fig. 1). Specifically, we define the retreat for each sector $i$ as

$$\Delta L_i(t) = \sum_{j \in i} f_j \, \Delta L_j(t) \Big/ \sum_{j \in i} f_j \tag{8}$$

where $f_j$ is the 2000-2010 mean observed ice flux (Enderlin et al., 2014; King et al., 2018) and the sum runs over all glaciers $j$ in ice-ocean sector $i$. This ensures that the largest glaciers are treated as the most important when generating a retreat projection per sector. Since we have $10^4$ retreat trajectories for each glacier (Fig. 4b), this procedure produces an ensemble of $10^4$ ice flux-weighted retreat trajectories for each ice-ocean sector. As expected, the median retreat of this ice flux-weighted ensemble is larger than the median retreat that would have been obtained by taking a simple mean over glaciers in a sector (Fig. 4c).

### 2.4.4 Low, medium and high scenarios

Given the large uncertainty associated with tidewater glacier response to climate forcing and the need to quantify uncertainties on future sea level contributions, it is desirable to provide a range of projected retreat that brackets the uncertainty associated with the retreat implementation. For each CMIP5 AOGCM we identify a low, medium and high retreat scenario (Fig. 1). From the ensemble of $10^4$ ice flux-weighted retreat trajectories for each ice-ocean sector, we define the medium retreat scenario as the trajectory with the median retreat at 2100 and the low and high retreat scenarios as the trajectories with the 25th and 75th percentile retreats at 2100 (Fig. 4c).

## 2.5 Submarine melt implementation

### 2.5.1 Extrapolation of ocean properties into fjords

In the submarine melt implementation, we account for the effects of fjord bathymetry and grounding line depth on the thermal forcing experienced by the glacier (Fig. 1). This is achieved by extrapolating the ocean property profiles (e.g. Fig. 3c) into fjords and below the present-day ice sheet by taking into account ocean bathymetry and subglacial topography in the same manner as Morlighem et al. (2019), based on the BedMachinev3 topography (Morlighem et al., 2017). Specifically, for each location in a fjord and beneath the present-day ice sheet, the deepest point that is openly connected to the wider ocean is determined; this depth is hereafter termed the effective depth. Water shallower than the effective depth is assumed to communicate directly with the open ocean and is assigned the temperature and salinity profile for the sector in question. Water deeper than the effective depth is not in direct communication with the open ocean because there is no continuous path to the open ocean that is not blocked by shallower bathymetry. Water deeper than the effective depth is, therefore, uniformly assigned a temperature and salinity equal to that at the effective depth.

An illustrative example is given for Sverdrup Glacier in NW Greenland and the adjacent ocean (Fig. 5). The fjord mouth has full-depth open communication with the ocean and is assigned unmodified ocean properties for the NW sector (yellow profiles in Figs. 5b-d). The bed topography at a point beneath the present day ice sheet reaches 600 m below sea level, but, assuming that the glacier had retreated past this point, would be separated from the open ocean by a sill at ∼350 m depth (Figs. 5a and b). By our extrapolation, this 600 m deep region is isolated from the warmest and saltiest water on the continental shelf. Thus the ocean properties in this deep region (red profiles in Figs. 5b-d) diverge from those at the fjord mouth below the height of the sill. This procedure is repeated for all fjords around the ice sheet, including below the present-day ice sheet, so that ocean conditions at calving fronts will be available to ice sheet models after calving fronts have retreated.

### 2.5.2 Calculation of ocean thermal forcing

In line with the more complex nature of the submarine melt implementation relative to the retreat implementation we use full, non-linear TEOS-10 routines (McDougall and Barker, 2011) to convert ocean property profiles to ocean thermal forcing profiles (Figs. 1 and 5d). Specifically, the CMIP5 quantities of depth, practical salinity and potential temperature are converted to pressure, absolute salinity and in-situ temperature using the 'gsw_p_from_z', 'gsw_SA_from_SP' and 'gsw_t_from_pt0' routines, respectively. A full three-dimensional, time-varying thermal forcing field $\text{TF}(x, y, z, t)$ is obtained as

$$\text{TF}(x, y, z, t) = T(x, y, z, t) - T_f(x, y, z, t) \tag{9}$$

where $T$ is the in-situ temperature and $T_f$ is the in-situ freezing point that depends on pressure and absolute salinity as defined by the 'gsw_t_freezing' routine. Lastly, we collapse the three-dimensional thermal forcing field to two-dimensions by considering only the value at the ocean bottom, so that the final thermal forcing field (TF) is defined at annual resolution on a 1 km $x$-$y$ grid covering Greenland (Fig. 6a). The motivation for using the ocean bottom value is that this is the thermal forcing experienced by the grounding line of a glacier if its calving front was located in the grid cell in question. Furthermore, plumes

upwell deep waters towards the fjord surface so that the temperature profile within the plume is well approximated by the value at the ocean bottom (Mankoff et al., 2016). We note that the submarine melt parameterisation is non-linear in TF (Eq. 2) so that annual mean melt is not equal to melt calculated from annual mean TF. The difference is, however, less than $1\%$ and it is, therefore, justified to use annual mean TF.

### 5  2.5.3  Assignation of runoff to drainage basins

The treatment of subglacial runoff is initially the same as for the retreat parameterisation. Once the time series of bias-corrected subglacial runoff has been obtained for each marine-terminating glacier (section 2.2), this runoff is distributed onto a 1 km $x$-$y$ grid by assigning the total runoff for each hydrological basin (Fig. 2b) to every grid point lying inside the basin (Figs. 1 and 6b). In this way, as a calving front retreats over the $x$-$y$ grid, the calving front submarine melt rate may be obtained by sampling

the ocean thermal forcing and subglacial runoff from the grid point at which the calving front is currently located. We assume that the hydrological drainage basins remain fixed in time at their present-day extent. Extending the runoff field beyond the present-day ice sheet is desirable to allow for potential calving front advance in the simulations, or to accommodate models whose initial ice extent is larger than observations. We choose to extrapolate subglacial runoff values beyond the present-day ice sheet by three 1 km grid cells using an iterative buffering approach. First, we sort the drainage basins by area from largest

to smallest. For each iteration, we buffer runoff values by one 1 km grid cell around each basin, starting with the largest basin and ending with the smallest basin. We fill only empty grid cells such that if a grid cell has already been populated by a runoff value from a larger basin, we do not overwrite that value. In this way, grid cells that are adjacent to two drainage basins are filled with runoff values from the larger basin. After the third iteration, we are left with a field of annual cumulative basin runoff values that have been extrapolated by three 1 km grid cells beyond the present-day ice sheet extent.

The submarine melt parameterisation Eq. (2) takes as input the subglacial runoff normalised by the submerged area of the calving front for each glacier. The submerged area will change over the course of the ice sheet model simulations as the termini retreat through fjords of various depths and widths. Since dynamically calculating the submerged area is difficult within an ice sheet model, we assume that the submerged area of each terminus remains constant at present-day values (c.f. Morlighem et al., 2019) but highlight this as an area for improvement in future efforts. The present-day submerged surface area is calculated

based on present-day calving front position and bed topography as defined by BedMachinev3 (Morlighem et al., 2017). Due to poor bed topography in some regions, which typically means unrealistically shallow topography in the region of a calving front, we impose a minimum submerged surface area of 0.2 km$^2$, equivalent to a glacier of width 2 km and grounding line depth 100 m.

### 2.5.4  Application of submarine melt parameterisation

Armed with both ocean thermal forcing and subglacial runoff fields defined at annual resolution on 1 km grids and with the submarine melt rate parameterisation Eq. (2), submarine melt rates may be estimated for the time period 1950-2100 and for each CMIP5 model (Fig. 1 and Table 1). While this defines a submarine melt rate on every grid cell where both ocean thermal forcing and runoff are defined (Fig. 6c), the intention is that the ice sheet model applies this submarine melt rate only when the

model has a calving front within this grid cell. In this way, the ice sheet models may apply a time-varying submarine melt rate to calving fronts around the ice sheet as these calving fronts retreat over the coming century.

## 3  Results

We here present the Greenland Ice Sheet ocean forcing arising from the choices and steps made in section 2. The intention is
to highlight temporal evolution of the forcing, together with spatial and model-to-model variability, as these factors will drive variability in sea level projections once implemented in an ice sheet model. The results are discussed with the same structure as section 2 and Fig. 1.

### 3.1  Future subglacial runoff

For both implementations, projected subglacial runoff is prescribed for each tidewater glacier using its hydrological drainage
basin. We visualise variability in runoff by considering runoff for the largest glacier by ice flux in each sector (Table S1; Fig. 3b), as these glaciers are likely to contribute the most to sea level over the coming century. These glaciers are Helheim (SE), Kangiata Nunata Sermia (SW), Kangerdlugssuaq (CE), Jakobshavn (CW), Daugaard-Jensen (NE), Kong Oscar (NW) and Humboldt (NO); note that in the retreat implementation, glaciers having permanent ice shelves have been excluded. Runoff shows high interannual variability and so we also plot and discuss smoothed curves.

In the MIROC5 RCP8.5 simulation, all glaciers show a significant increase in runoff by 2100, with most of the increase occurring after 2050 (Fig. 7a). Jakobshavn (CW) and Humboldt (NO) show the largest absolute increase in runoff, with Daugaard-Jensen (NE) and Kong Oscar (NW) having the smallest runoff anomaly (Fig. 7a). A different picture of spatial variability emerges when considering the relative runoff anomaly (Fig. 7b). In this case it is Kong Oscar (NW) that stands out, with JJA runoff in 2100 a factor of 8 larger than during the 1995-2014 baseline period. Kangiata Nunata Sermia (SW) also
experiences a large relative increase in runoff, while Daugaard-Jensen (NE) sees the smallest, amounting to only a factor 2.5 larger than in 1995-2014. Equivalent plots for all other CMIP5 AOGCMs are shown in Figs. S4 and S5 but show very similar spatial variability to MIROC5.

Lastly, we consider model-to-model variability in projected runoff by averaging over the largest glaciers by sector (Fig. 7c). The only RCP2.6 scenario considered shows a moderate increase in runoff until 2050 before a return to present-day values by
2100 (Fig. 7c). All RCP8.5 simulations exhibit a similar temporal evolution and show a significant increase in runoff during the coming century. HadGEM2-ES has the highest runoff at $\sim$2000 m$^3$s$^{-1}$ in 2100, with IPSL-CM5A and MIROC5 giving similar results. NorESM1-M and ACCESS1-3 have medium runoff and CSIRO-Mk3-6-0 has the lowest runoff at $\sim$1150 m$^3$s$^{-1}$ by 2100. The multi-model spread in runoff anomaly at 2100 is $\sim$850 m$^3$s$^{-1}$, around 50% of the multi-model mean of $\sim$1650 m$^3$s$^{-1}$. Relative to the 1995-2014 mean of 440 m$^3$s$^{-1}$, these projections suggest an average increase in runoff by a factor 2.5-
4.5 this century. The model-to-model variability is as would be expected from the ISMIP6 CMIP5 model evaluation exercise (Barthel et al., 2019).

## 3.2 Future ocean thermal forcing

We present ocean results based on the sector-averaged, depth-averaged time series derived for the retreat implementation (section 2.4.1). While the submarine melt implementation differs by retaining depth variability and through the extrapolation of properties into fjords, the depth-averaged values from the retreat implementation remain a reliable indicator of what the ocean does.

There is significant regional variability in projected ocean warming in the MIROC5 RCP8.5 simulation (Fig. 8a). The NE sector stands out with a thermal forcing increase of nearly 5°C, while all other sectors exhibit an increase of between 1 and 3°C. Ocean warming in the NE sector amounts to an increase of 150% in thermal forcing relative to the 1995-2014 baseline period (Fig. 8b). The SE and SW sectors see the smallest relative increase amounting to only ∼20%. We do note, however, that regional ocean warming differs substantially across CMIP5 AOGCMs (Yin et al., 2011; Barthel et al., 2019, Figs. S6 and S7). The NE sector sees the most warming in MIROC5, HadGEM2-ES and IPSL-CM5A-MR, but the CW and NW regions see equivalent or greater warming in the other three RCP8.5 models. It is also interesting to note that the relative increase in runoff (Fig. 7b) is much larger than the relative increase in ocean thermal forcing (Fig. 8b).

We consider ocean warming at the ice sheet scale by taking a mean over the 7 sectors for each CMIP5 AOGCM (Fig. 8c). For MIROC5 RCP2.6, there is moderate warming of nearly half a degree which persists until the end of the century. This is mostly driven by significant warming in the CW and NW sectors (Fig. S6a) that exceeds warming in these sectors in some of the RCP8.5 simulations (Figs. S6d and S6f). Given the large inter-model variability in ocean warming, this warming feature is likely to be specific to MIROC5 rather than being more broadly representative of RCP2.6 simulations. Among the RCP8.5 simulations, CSIRO-Mk3-6-0 shows the most warming by 2100, reaching 2.8°C above the present-day value. HadGEM2-ES shows the least warming, reaching 1.9°C by 2100. The multi-model spread in thermal forcing anomaly by 2100 is 0.9°C, around 35% of the multi-model mean of ∼2.4°C. Relative to the 1995-2014 baseline value of 4.6°C, thermal forcing is projected to increase by a factor 0.4-0.6 this century under RCP8.5 (Fig. 8c).

## 3.3 Retreat implementation forcing

Projected sector retreat combines the runoff anomaly per glacier (section 3.1), the thermal forcing anomaly per sector (section 3.2) and the ice flux of all glaciers in the sector (section 2.4.3). Thus, sector-to-sector variability in projected retreat arises due to both variability in regional climate and differences in the population of glaciers in each sector.

For the MIROC5 RCP8.5 simulation, the SW sector has the largest retreat (Fig. 9a) because it has a small number of glaciers (Table S1) each experiencing a large increase in subglacial runoff (Figs. 7a-b). The projected retreat for the CW sector is also high (Fig. 9a), partly due to large projected retreat for Jakobshavn, which dominates the sector-average retreat because it alone accounts for around half of the present-day ice flux in the CW sector (Table S1). Projected retreat is smallest for the NW and NO sectors (Fig. 9a) because these sectors comprise a large population of smaller glaciers (Table S1) and experience the least absolute increase in subglacial runoff (Fig. 7a). Fig. S8 shows equivalent plots to Fig. 9a for all other CMIP5 AOGCMs, in which the spatial patterns of retreat are similar in almost all models with large projected retreat for SW and CW and smaller

retreat for NW and NO. Note that Fig. 9a shows only the medium retreat case for each sector; low and high projections are plotted in Fig. S9.

To provide an ice sheet-wide view of retreat per CMIP5 AOGCM, we combine the sector-by-sector projections (e.g. Fig. 9a) into an ice sheet projection by weighting according to the present-day ice flux (Table S1). The resulting projections (Fig. 9b) are not used to force the ice sheet models (the ice sheet models are forced by the sector-by-sector projections) but they do illustrate multi-model variability in projected retreat. The RCP2.6 simulation considered shows moderate retreat of ∼2 km until 2050 and then a stabilisation of terminus positions (Fig. 9b). The retreat is largely driven by significant ocean warming in the CW and NW sectors (Figs. S6a and S8a).

The RCP8.5 projections show ∼15 km of retreat by 2100. The retreat rate generally increases throughout the century, so that ∼4 km of retreat occurs before 2050 and ∼11 km between 2050 and 2100. The multi-model spread in retreat by 2100 is only 2 km, or 15% of the multi-model mean. The largest retreat is projected using CSIRO-Mk3-6-0 and the least using HadGEM2-ES, although all models are similar. In contrast, the spread in projections resulting from the low and high retreat cases for a given model is generally large. For the MIROC5 RCP8.5 projections, the difference between the low and high retreat cases at 2100 is 14 km, much larger than the multi-model spread (Fig. 9b). The same is true for the low and high cases in all other RCP8.5 models (not shown).

## 3.4 Submarine melt implementation forcing

Projections of submarine melt rates are obtained by combining ocean thermal forcing, runoff accumulated over each glacier's subglacial drainage basin and a calving front submerged area (Eq. (2)). To illustrate the results, we show melt rates for the glacier with the largest ice flux in each region (Fig. 10; Table S1). These projections do not take into account the motion of glacier termini and, thus, isolate the change in melt rates due solely to changes in future atmospheric and ocean forcing.

Submarine melt rates increase over the projection timespan (2015-2100) under all RCP8.5 scenarios for all 7 glaciers, although the magnitude and timing of the increase varies by location and by CMIP5 AOGCM (Fig. 10). At Humboldt Glacier (NO), little increase is seen until 2060, after which the models diverge with a range of 0.5 to 2 $m d^{-1}$ projected melt rate in 2100 (Fig. 10b). In the NW, NE and central regions (Jakobshavn, Kangerdlussuaq, Kong Oscar and Daugaard-Jensen), melt rates increase soon after 2015 and are up to 5 times larger in 2100 relative to the 1995-2014 baseline period (Figs. 10c-f). In the south (Kangiata Nunata Sermia and Helheim), melt rates double or triple by 2100 under RCP8.5 (Figs. 10g-h). There is significant spread in projected melt rates in 2100 for the RCP8.5 scenarios, typically amounting to 25-50% of the multi-model mean but substantially more for Humboldt Glacier. When considering a mean over the 7 glaciers, the multi-model spread under RCP8.5 is much smaller than at individual glaciers, with the mean melt rate increasing from ∼2 $m d^{-1}$ in the present day to ∼6 $m d^{-1}$ in 2100 (Fig. 10a). Under the RCP2.6 scenario, melt rates show only moderate increases until around 2050, followed by stabilisation or decrease (Fig. 10). Projected RCP2.6 melt rates in 2100 are lower than the present day for Kangiata Nunata Sermia and Helheim (Figs. 10g-h). In general, RCP2.6 melt rates do not depart significantly from RCP8.5 melt rates until around 2050.

A similar picture emerges when a larger population of 125 glaciers is considered. Fig. 11 shows histograms of the relative increase in submarine melt rate between a twenty year period the end of the century (2081-2100) and the present day (1995-2014) under all 6 of the RCP8.5 models considered. For example, since we consider 58 glaciers in NW Greenland, Fig. 11a has a total count of 58 x 6 = 348. In SE and SW Greenland, melt rates increase by at most 170%. These regions already experience a warm ocean and atmosphere in the present day and so large increases in absolute melt rate (Figs. 10g-h) appear as smaller relative increases in submarine melting. Moving north, CE, CW and NW Greenland experience increases up to ~400% while the NE and NO sectors have the largest relative increases in melting, reaching over 1000%. These northerly regions have a particularly cold ocean in the present day and currently experience very little submarine melting (e.g. Fig. 10b). Thus any increase in absolute melt rate can constitute a very large relative increase.

The spread in relative melt rate increase within regions (Fig. 11) arises from a number of factors. The glaciers in each region have diverse grounding line depths, submerged in fjords with differing sill depths. Thus, glaciers with deep grounding lines that are directly exposed to the ocean are responding to different water masses than glaciers that are grounded in shallow water or protected from the ocean by shallow sills. If these water masses evolve differently over the coming century then adjacent glaciers may experience very different ocean forcing even within the same CMIP5 AOGCM. A second source of variability is that from the 6 CMIP5 AOGCMs themselves, which can differ substantially on the evolution of ocean temperature within a given sector (Fig. S7).

## 4 Discussion

### 4.1 Retreat and submarine melt implementations

The two implementations have distinct advantages and disadvantages. The retreat implementation has the advantage of being accessible to all ISMIP6 ice sheet models (Goelzer et al., 2020) and has been empirically validated by tuning to match observed glacier retreat over the past 60 years (Slater et al., 2019). In addition, it replaces the need for a representation of calving, the parameterisation of which remains a large source of uncertainty (Benn et al., 2017). On the other hand, the retreat implementation does parameterise terminus position in a very constraining manner: it does not allow for modeled ice dynamics to influence the terminus position and it takes no account of bed topography, which is known to be an important factor in determining the response of an individual glacier to an ocean perturbation (e.g. Catania et al., 2018). These issues motivate the second proposed implementation.

The submarine melt implementation places less constraints on the interaction between the ocean and ice sheet by specifying only the submarine melt rate (or more precisely, the subglacial runoff, ocean temperature and a parameterisation to combine these quantities to estimate submarine melt rate). The representation of calving and its possible coupling to submarine melting is left to the ice sheet model. This implementation has the advantage that the important interactions between submarine melting, calving, ice dynamics and bed topography can be resolved by the model (e.g. Aschwanden et al., 2019; Morlighem et al., 2019). The disadvantages are that there is large uncertainty in the submarine melt rates obtained from the parameterisation and we

still lack a good understanding of - and parameterisation for - calving. Furthermore, the submarine melt implementation may be considerably more computationally expensive and technically challenging to implement than the retreat implementation.

## 4.2 Variability in projections

The projected relative increase in subglacial runoff (factor 2.5-4.5, Fig. 7) is much higher than for ocean thermal forcing (factor 0.4-0.6, Fig. 8) for all models under an RCP8.5 scenario. Yet both forcings contribute significantly to the retreat and submarine melt rate projections due to the form of the retreat and submarine melt parameterisations (Eqs. (1) and (2)). The subglacial runoff $Q$ appears sub-linearly in these parameterisations, while the thermal forcing (TF) appears approximately linearly, so that the impact of increasing thermal forcing on projected retreat and submarine melting is larger than the impact of increasing runoff by an equivalent relative amount.

There also appears to be some compensation occurring between atmosphere and ocean in the 6 AOGCMs we have considered. The multi-model spread by 2100 in projected subglacial runoff is ∼50% and in thermal forcing is ∼35%, but the spread in projected retreat and submarine melting is only ∼15% (Figs. 7c, 8c, 9b and 10a). The model that has the most ocean warming (CSIRO-Mk3-6-0) has the least runoff increase and the model that has the least ocean warming (HadGEM2-ES) has the most runoff increase (Figs. 7c and 8c). Due to the form of the retreat and submarine melt parameterisations (Eqs. (1) and (2)), the atmosphere and ocean projections can compensate each other, reducing the multi-model spread in the retreat and submarine melt projections. Coupled with the large uncertainty in the linear coefficient $\kappa$ appearing in the retreat parameterisation (Slater et al., 2019), the spread in projected retreat due to the low and high retreat cases (section 2.4.4) is, therefore, much larger than the spread in projected retreat or submarine melting due to AOGCM selection (Figs. 9b and 10a). It can therefore be expected that the spread in sea level projections arising from the use of the low and high retreat scenarios will be larger than from the medium retreat or submarine melt rate scenarios forced by different CMIP5 AOGCMs. In terms of sampling uncertainty on future sea level within the implementations presented here, it may be more beneficial to prioritise ice sheet simulations sampling uncertainty in coefficients in the parameterisations rather than considering additional AOGCMs. We note that this may not be true for a different ocean forcing implementation and that we have only considered 6 CMIP5 AOGCMs in this study (Table 1) that are a selected subset of the larger CMIP5 ensemble (Barthel et al., 2019). It is possible that use of other CMIP5 AOGCMs would lead to a greater spread in projected retreat and submarine melting.

Examination of the projected submarine melt rates also suggests the possibility for sector-by-sector compensation. For example, CSIRO-Mk3-6-0 has the highest projected melt rates of any RCP8.5 model at Kong Oscar, Jakobshavn and Helheim but is close to the lowest projected melt rates at Humboldt, Daugaard-Jensen and Kangiata Nunata Sermia (Fig. 10). There is no individual CMIP5 AOGCM that gives high melt rates in every single sector or at every single glacier; rather a model that gives high melt rates in a certain sector often gives lower melt rates in another sector. As a result, taking a mean of the projected RCP8.5 melt rates over 7 large glaciers gives trajectories that lie within a narrow envelope (Fig. 10a). Once again, this may act to reduce the spread in the projected sea level from ice sheet models forced by these melt rates. The response of individual sectors or glaciers may differ substantially between CMIP5 AOGCMs, but Fig. 10 suggests that glaciers and sectors may compensate one another, leading to a similar sea level contribution from the full ice sheet under each CMIP5 AOGCM.

The dynamic sea level contribution is not, however, directly related to the magnitude of retreat or submarine melt rate. For example, although the SW sector has the largest projected retreat, it contains relatively few tidewater glaciers and these glaciers currently account for <4% of Greenland's ice discharge (Table S1). It is therefore unlikely to be a major source of dynamic sea level contribution in the future. In contrast, the NW region has the smallest projected retreat but has a large
number of tidewater glaciers that currently account for ∼20% of Greenland's ice flux (Table S1) and is much more likely to be a significant dynamic contributor to sea level. Within the submarine melt implementation there is also the possibility for non-linear or threshold response of glaciers to submarine melting, where small changes in forcing may result in large excursions in terminus position and mass loss (Morlighem et al., 2019).

## 4.3   Impact of bias corrections

In order to provide continuous ocean forcing from the present day into the future and to ensure we sample uncertainty in future climate projection rather than representation of the present day, we bias-corrected the subglacial runoff (section 2.2.3) and ocean thermal forcing (section 2.3.3). Due to the non-linearity of the retreat and submarine melt parameterisations, the bias corrections do impact the projected retreat and submarine melt rate. Where there exists uncertainty in the present-day quantities, for example the ocean thermal forcing in CW, NW and NO Greenland, this leads to uncertainty in the projections.

Compared to the situation in which no bias correction is performed, the bias correction can change RCP8.5 projected retreat by 2100 by up to a few kilometers, or around 0 to 20% of the typical retreat by 2100 of 15 km (Figs. 9b and S10). The bias correction is equally likely to increase or decrease the projected retreat (Fig. S10). There are a few instances where the impact of the bias correction is larger, for example in NorESM1-M the bias correction decreases projected retreat by 36% in SE Greenland and increases it by 20% in CW Greenland (Fig. S10). These follow from the large ocean thermal forcing bias corrections applied
to this model (Fig. S3). The bias corrections can, therefore, contribute to sector-by-sector differences in retreat projections but do not overall increase or decrease projected retreat. Since the retreat and submarine melt parameterisations have a similar form, the impact of the bias correction on submarine melting will be similar.

## 4.4   Missing processes and priorities for future improvement

Due to the complexity and timescale of the exercise we have had to make a number of simplifications of complex processes in
order to deliver the ocean forcing to the ice sheet modeling groups. One key simplification is our treatment of the ocean thermal forcing experienced by tidewater glaciers. Since the CMIP AOGCMs do not resolve Greenland's fjords, we have had to bridge the gap between the continental shelf and calving fronts. In the retreat parameterisation, the ocean thermal forcing applied to glaciers is a spatially- and depth-averaged value from the continental shelf. Thus, we have neglected spatial gradients in ocean temperatures within the chosen sectors (Slater et al., 2019), the processes responsible for transporting and transforming ocean
waters between the shelf and calving front (Motyka et al., 2003; Straneo et al., 2010; Mortensen et al., 2011; Jackson et al., 2014; Gladish et al., 2015) and the diverse grounding line and sill depths of glaciers and fjords in Greenland (Morlighem et al., 2017). We do note that the retreat parameterisation Eq. (1) was tuned based on observations from 1960 to present using the same definition of ocean thermal forcing and so, to some extent, all of these processes will have fed into the empirical tuning.

This definition of ocean thermal forcing nevertheless neglects much of the individuality of glacier-fjord systems, essentially linking groups of glaciers to large-scale ocean changes only.

In the submarine melt implementation, the effect of sills and grounding line depth is taken into account by retaining the depth-variability of ocean conditions and extrapolating these properties into fjords based on the bathymetry. Certainly, the presence of sills is known to modify fjord water properties substantially by blocking access of dense waters to the calving front (Gladish et al., 2015), but this extrapolation remains a simplification due to vertical mixing within fjords (e.g. Inall et al., 2014) and because periodic dense inflows over sills have been observed in Greenland (Mortensen et al., 2011). Therefore, both the retreat and submarine melt implementations would be improved with methods to quantify water mass transformation between the shelf and calving fronts. Such methods might take the form of very high-resolution regional ocean modeling or, perhaps more practically for efforts such as ISMIP6, simple parameterisations or fjord box models. Knowledge of fjord and shelf bathymetry is a prerequisite for these improvements but is currently incomplete and is, therefore, an additional priority.

Both implementations also assume that submarine melting is the primary climate forcing experienced by the calving fronts of tidewater glaciers. This assumption derives from the literature consensus on the important role played by submarine melting in the recent retreat of tidewater glaciers in Greenland (Holland et al., 2008; Straneo and Heimbach, 2013; Fried et al., 2015; Cowton et al., 2018), yet other processes may play a substantial role. In particular, the buttressing provided to glaciers by ice mélange may be sufficient to suppress calving (Amundson et al., 2010; Robel, 2017), has been implicated in rapid glacier retreat (Christoffersen et al., 2012; Moon et al., 2015; Bevan et al., 2019) and is found to be more influential than submarine melt in some models (Krug et al., 2015; Todd et al., 2018). Future ice sheet-ocean forcing efforts might, therefore, look to incorporate the impact of ice mélange buttressing.

Once submarine melting is assumed to be the primary ocean forcing, it must be parameterised, as has been done in Eqs. (1) and (2) for the retreat and submarine melt implementations, respectively. The form of both parameterisations derives from the physics of plumes, which, aside from the submarine melt they induce, are relatively well understood from theory, laboratory and observational work (Morton et al., 1956; Jenkins, 2011; Jackson et al., 2017). Observations of submarine melting with which to constrain key constants in melt parameterisations are, however, severely lacking. Our first direct observations of submarine melting were obtained very recently in Alaska (Jackson et al., 2019; Sutherland et al., 2019) and suggest we may currently be underestimating submarine melt rates, especially outside of plumes. For the retreat implementation, uncertainty in melt parameterisations is less of an issue because the parameterisation assumes proportionality between glacier retreat and submarine melt rate, and since glacier retreat is easily observable, we have good observations to tune the linear coefficient $\kappa$ (Slater et al., 2019). This is not the case for the submarine melt implementation, though ice sheet models typically do a spin-up simulation in which they tune their model to try to match present-day ice sheet extent, which may go some way to reducing their sensitivity to uncertainty in the melt parameterisation. It is clear, however, that observations of submarine melting and further work building on Sutherland et al. (2019) would be valuable for reducing uncertainties on sea level contribution in efforts beyond ISMIP6.

## 5 Summary

The Ice Sheet Model Intercomparison Project for CMIP6 (ISMIP6) constitutes the primary community effort to produce ice sheet sea level projections for the next Intergovernmental Panel on Climate Change Assessment Report (IPCC AR6). ISMIP6 is the first effort to develop a multi-model ensemble of Greenland Ice Sheet models forced by ocean boundary conditions derived from CMIP AOGCMs. Such a strategy is demanding to design due to the evolving nature of our process understanding and ice sheet model technical capabilities. With these challenges in mind, we have proposed two ocean forcing strategies, called the retreat implementation and the submarine melt implementation. By combining these strategies with projected climate from selected CMIP AOGCMs, we have derived ocean boundary conditions for Greenland Ice Sheet models to run 21st century projections (Goelzer et al., 2020).

In the retreat implementation, retreat is projected using a process-motivated but empirically-calibrated parameterisation that combines subglacial runoff and ocean thermal forcing to estimate tidewater glacier retreat (Slater et al., 2019). Retreat is projected for each individual tidewater glacier but for simplicity is applied to the ice sheet homogeneously within each of 7 sectors. Under a high greenhouse gas emissions RCP8.5 scenario, projected retreat that will be applied to the ice sheet models amounts to around 15 km by 2100 with a range of 10-25 km in low and high scenarios. Under a low emissions RCP2.6 scenario, retreat of only ∼1 km will be prescribed. In the submarine melt implementation, fields of subglacial discharge and ocean thermal forcing covering Greenland are provided, together with a recommended parameterisation that may be used to estimate submarine melt rate wherever a calving front is located. Under RCP8.5, projected melt rates in 2100 are a factor ∼3 higher than the present day but remain relatively constant under RCP2.6. The sea level contributions resulting from these two implementations will be determined by the modeled dynamic response to these forcings.

The proposed implementations are driven by process understanding but are also pragmatic and have necessarily neglected certain processes or made use of poorly-constrained parameterisations. Foremost amongst these are fjord processes and the transformation of ocean waters between the continental shelf and glacier calving front and the parameterisation of submarine melting. These issues are to some extent ameliorated through tuning, both in the described implementation and at the level of the ice sheet model. Nevertheless, research constraining submarine melt parameterisations and calving laws and developing simple methods for quantifying fjord transformation of ocean waters should remain a high priority for reducing uncertainty on the future sea level contribution of the Greenland Ice Sheet.

| MODEL | SCENARIO |
|---|---|
| MIROC5 | RCP2.6 & RCP8.5 |
| NorESM1-M | RCP8.5 |
| HadGEM2-ES | RCP8.5 |
| CSIRO-Mk3-6-0 | RCP8.5 |
| IPSL-CM5A-MR | RCP8.5 |
| ACCESS1-3 | RCP8.5 |

**Table 1.** CMIP5 AOGCMs and scenarios considered.

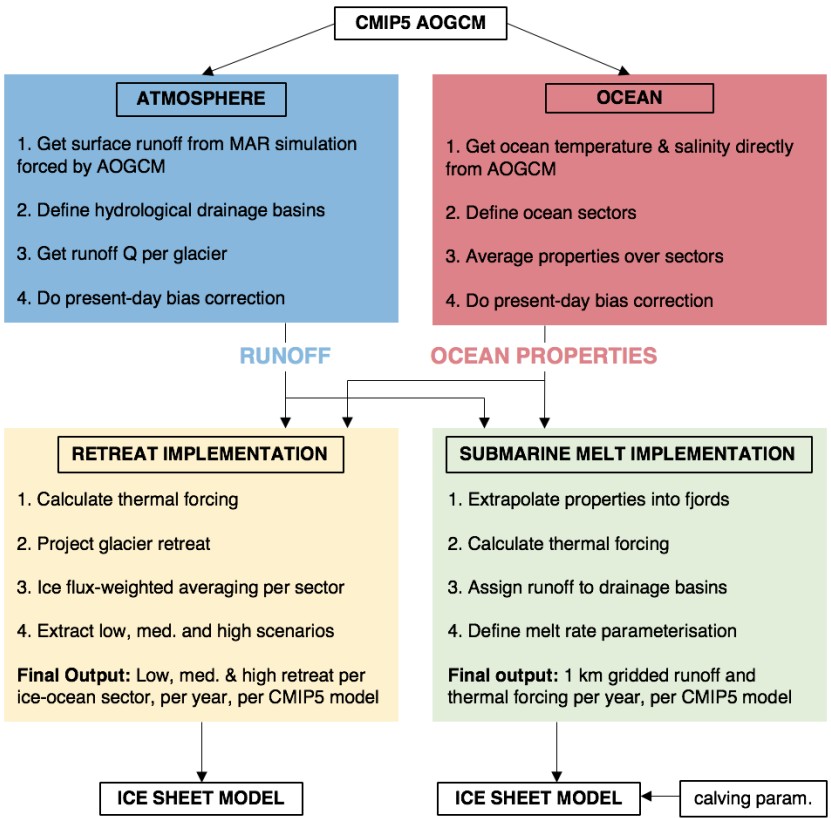

**Figure 1.** Schematic of proposed approach to use CMIP5 AOGCM output (top) to force Greenland Ice Sheet models (bottom) under the retreat and submarine melt implementations described in the text. The colored boxes describe the methodology and analysis performed in this paper. Note that the process would be identical for CMIP6 models.

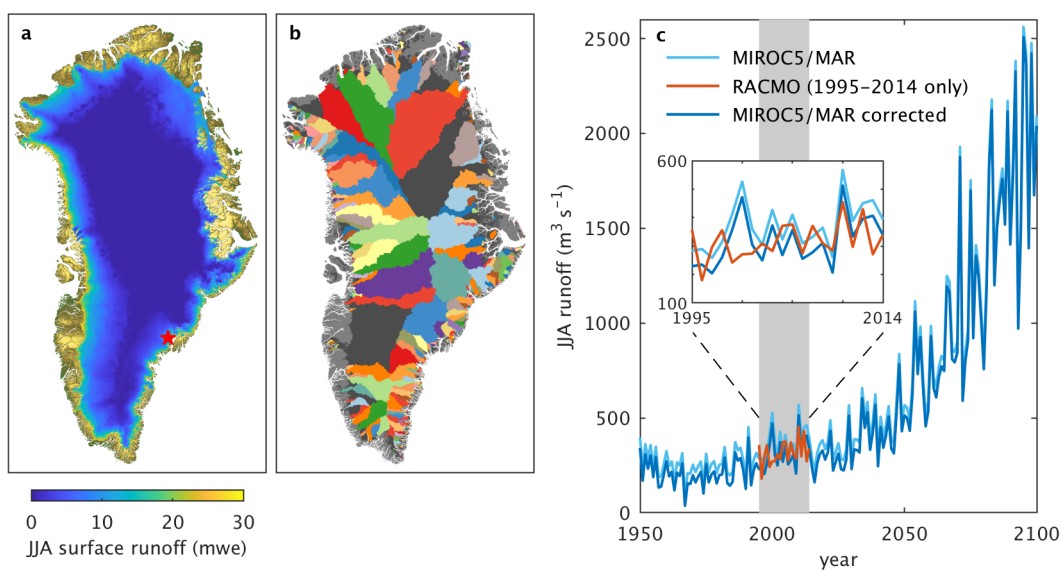

**Figure 2.** Illustration of atmospheric processing for the MIROC5 RCP8.5 scenario. (a) Simulated June-July-August (JJA) surface runoff in 2100 in the regional climate model MAR3.9.6 forced at its boundaries by MIROC5. (b) Tidewater glacier drainage basins delineated based on the hydropotential defined in Eq. (3). (c) JJA runoff time series for Helheim Glacier in SE Greenland (location shown as red star on (a)). The vertical grey shading shows the 1995-2014 present-day time period used for the bias correction. The raw MAR output is in light blue, RACMO during the present-day period is in red and the bias-corrected MAR output is in dark blue. The inset shows a zoom-in for the present-day time period.

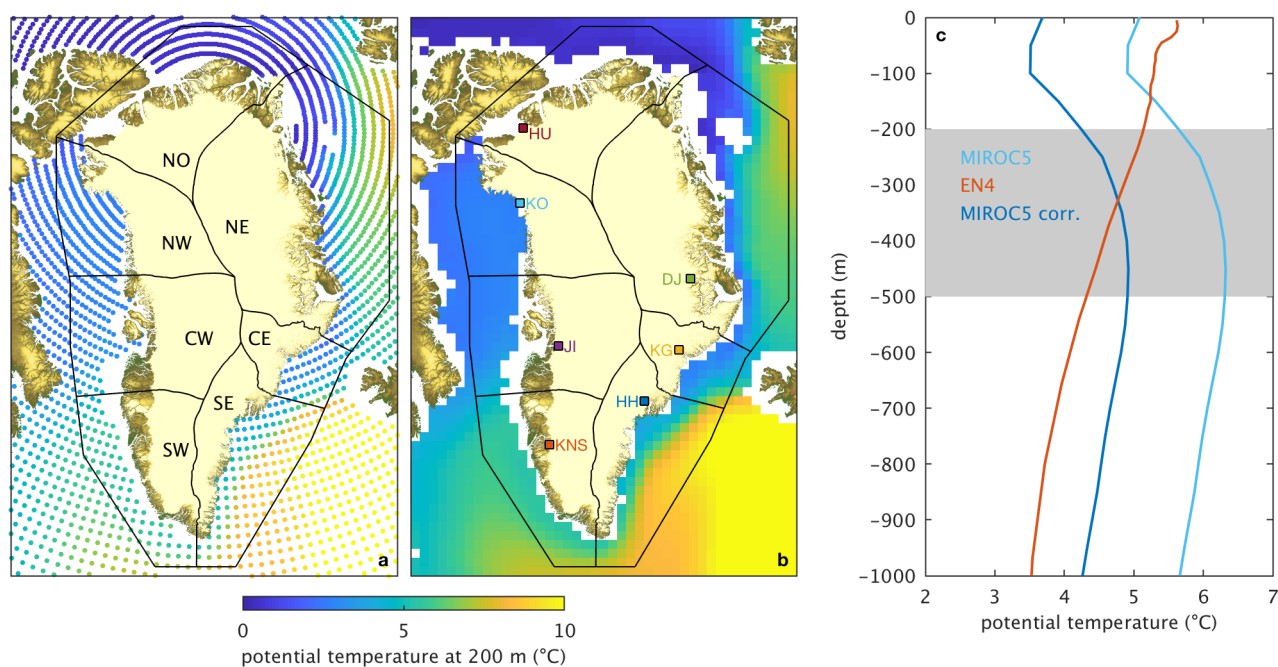

**Figure 3.** Illustration of ocean processing for the model MIROC5 in an RCP8.5 scenario. (a) Modeled annual mean potential temperature at 200 m in the year 2100, plotted at the 1.4° resolution of the climate model. The seven ice-ocean sectors over which properties are averaged are also shown and labeled. (b) The same variable, gridded at 50 km resolution for spatial averaging. Also shown are the largest glaciers by ice flux in each sector: HH (Helheim), KNS (Kangiata Nunata Sermia), KG (Kangerdlugssuaq), JI (Jakobshavn Isbrae), DJ (Daugaard-Jensen), KO (Kong Oscar) and HU (Humboldt). (c) Ocean temperature bias correction for the SE sector. All three profiles are temporal averages over the 1995-2014 present-day period. The raw MIROC5 output (light blue) is compared to the observational (EN4) profile (red) and is bias-corrected (dark blue), so that the depth-average over the 200-500 m range (shaded grey) agrees with EN4.

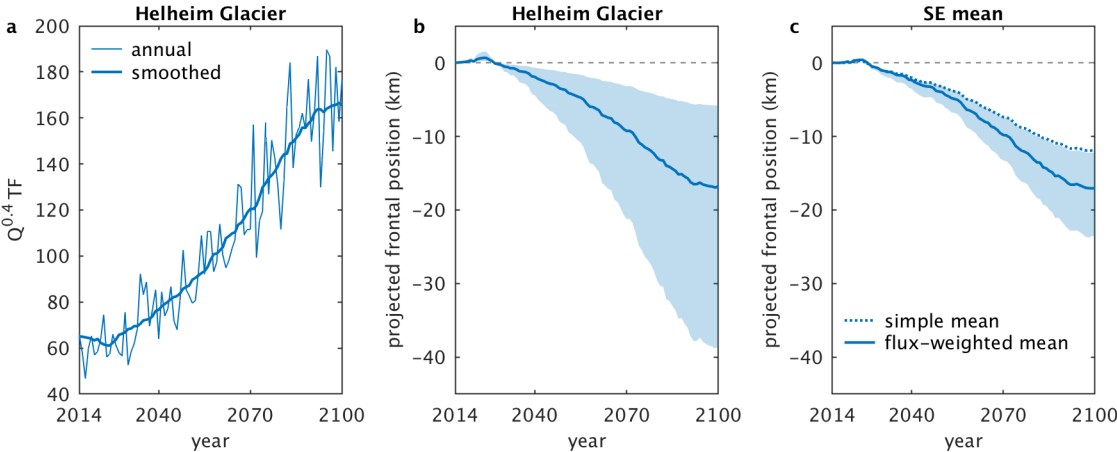

**Figure 4.** Illustration of retreat implementation processing for MIROC5 under an RCP8.5 scenario. (a) Time series of $Q^{0.4}$ TF for Helheim Glacier in SE Greenland, showing annual and 20-year centered mean smoothed values. (b) Projected retreat for Helheim Glacier; solid line is the median retreat while the shading denotes the interquartile range of all $10^4$ derived retreat trajectories. (c) Projected retreat for the SE ice-ocean sector. The dotted line shows the median of the trajectories obtained by taking a simple mean over glaciers, while the solid line and shading show the median and interquartile range of the trajectories obtained by taking an ice flux-weighted mean over glaciers.

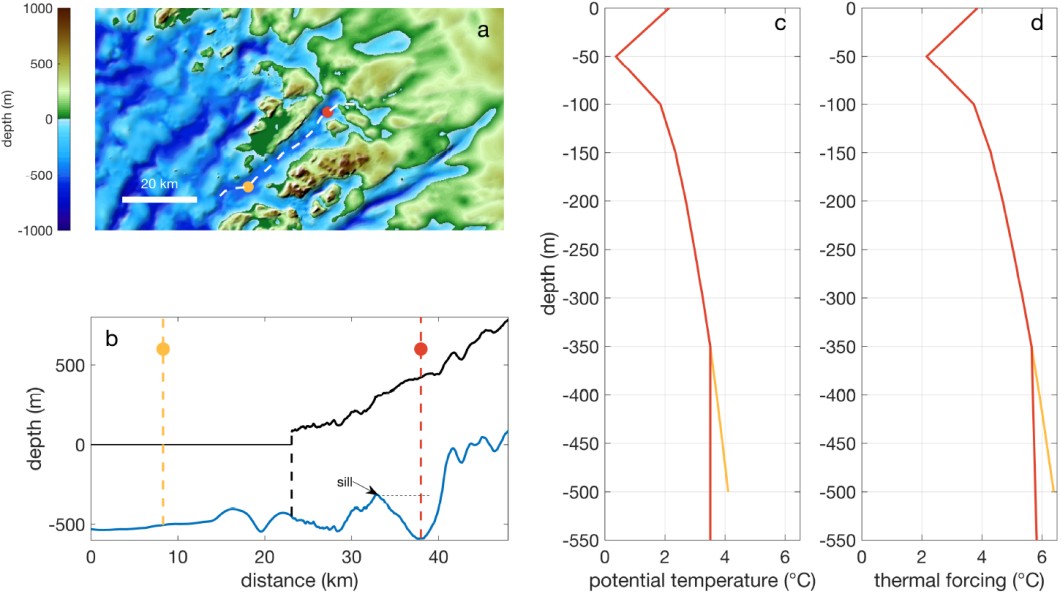

**Figure 5.** Illustration of ocean property extrapolation for Sverdrup Glacier and fjord, NW Greenland. (a) Overview of regional topography. The white dashed line shows an along-fjord transect, the yellow point is in the fjord and the red point is below the present-day ice sheet. (b) Bathymetry/subglacial topography (blue) and current ice sheet elevation (black) along the flowline shown as the white dashed line in (a). The yellow and red dashed lines correspond to the locations in (a). (c) Potential temperature profiles and (d) thermal forcing profiles at the locations shown in yellow and red in (a) and (b).

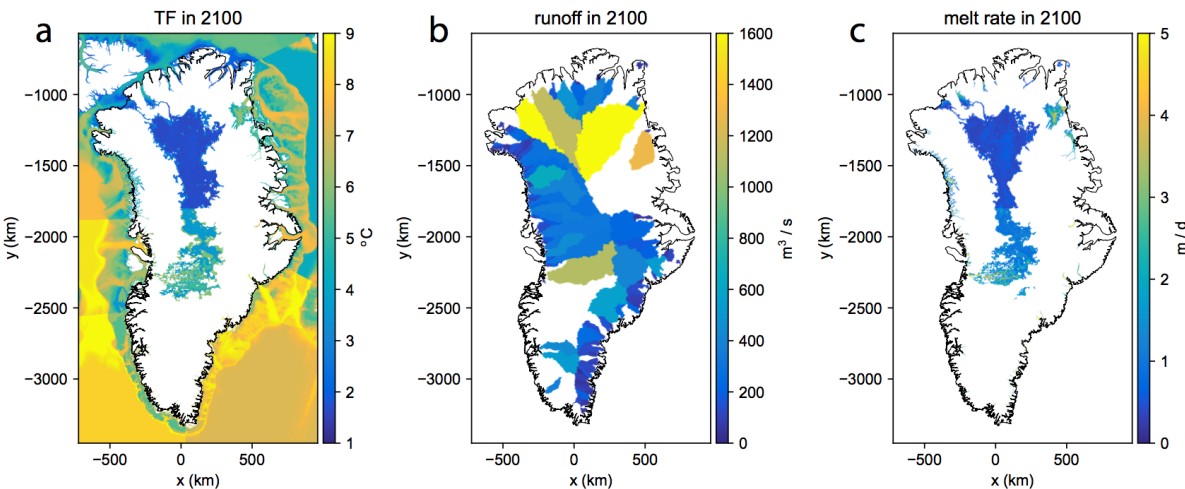

**Figure 6.** Example of forcing fields in 2100 in the submarine melt implementation, using MIROC5 under an RCP8.5 scenario. (a) Ocean thermal forcing, (b) subglacial runoff and (c) submarine melt rate calculated using the parameterisation in Eq. (2). Note that the thermal forcing and melt rate values in the ice sheet interior are included only to show that the submarine melt implementation defines melt rate everywhere that is below sea level and connected to the ocean. An ice sheet model would only apply these melt rates if the ice sheet margin retreats into the interior, which is unlikely by 2100. Also note that runoff values are only plotted for marine-terminating hydrological basins.

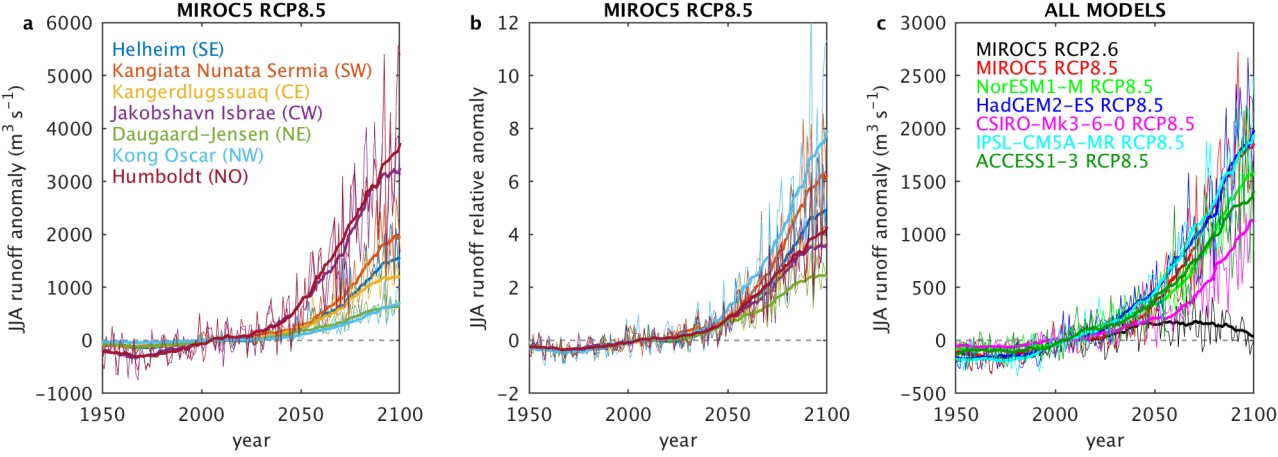

**Figure 7.** Projected subglacial runoff. For clarity, annual values are plotted as thin lines and 20-year running means are plotted as thicker lines. (a) Absolute runoff anomaly (difference from the 1995-2014 mean) by sector in the MIROC5 RCP8.5 simulation. For each sector, the runoff anomaly for the largest glacier by ice flux in that sector is plotted (locations in Fig. 3b). (b) Runoff anomaly normalised by the present-day value for the same glaciers (absolute anomaly divided by the 1995-2014 mean). The legend is as for (a). (c) Representative runoff anomaly per CMIP AOGCM to illustrate model spread. The representative runoff anomaly is calculated as the mean over the 7 glaciers shown in (a). Full plots for all sectors and models may be found in Figs. S4 and S5.

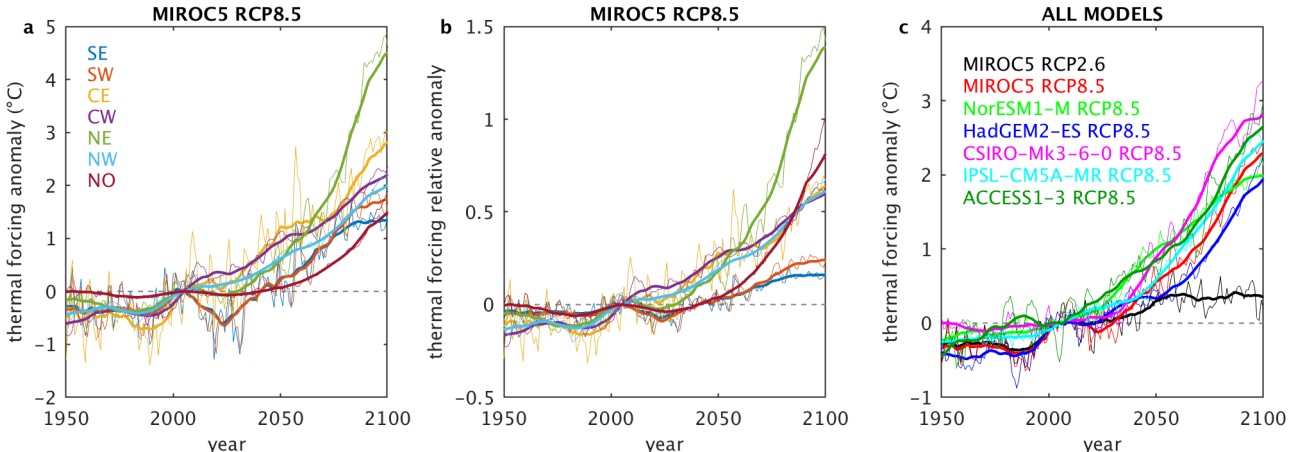

**Figure 8.** Projected 200-500 m ocean thermal forcing. (a) projected absolute thermal forcing anomaly per ice-ocean sector in the MIROC5 RCP8.5 simulation. (b) Thermal forcing anomaly normalised by the present-day value. The legend is as for (a). (c) Mean thermal forcing over the 7 ice-ocean sectors for each CMIP AOGCM. Full plots for all sectors and all models may be found in Figs. S6 and S7.

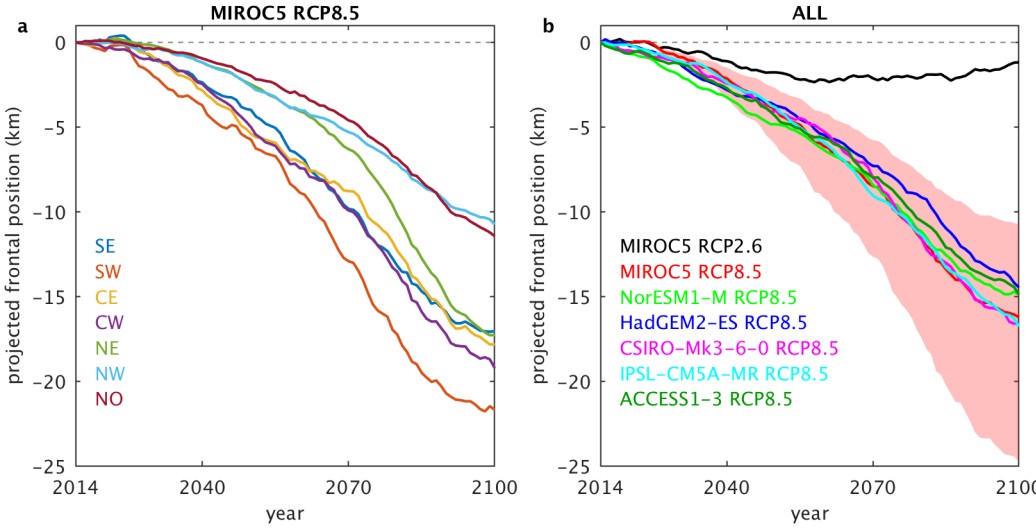

**Figure 9.** Projected tidewater glacier frontal position for forcing of Greenland Ice Sheet models. (a) Sector-by-sector retreat from the MIROC5 RCP8.5 simulation, showing only the medium retreat. (b) Retreat in all CMIP AOGCMs considered, where the sectors in (a) are combined according to their present-day relative ice flux (Table S1). Also shown in the shading is the low and high retreat projections for MIROC5 RCP8.5. Note that the ice sheet models are forced on a sector-by-sector basis, so the projections in (b) are not used to force any models but are included to give a sense of the multi-model variability. See Fig. S8 for full plots of all projections.

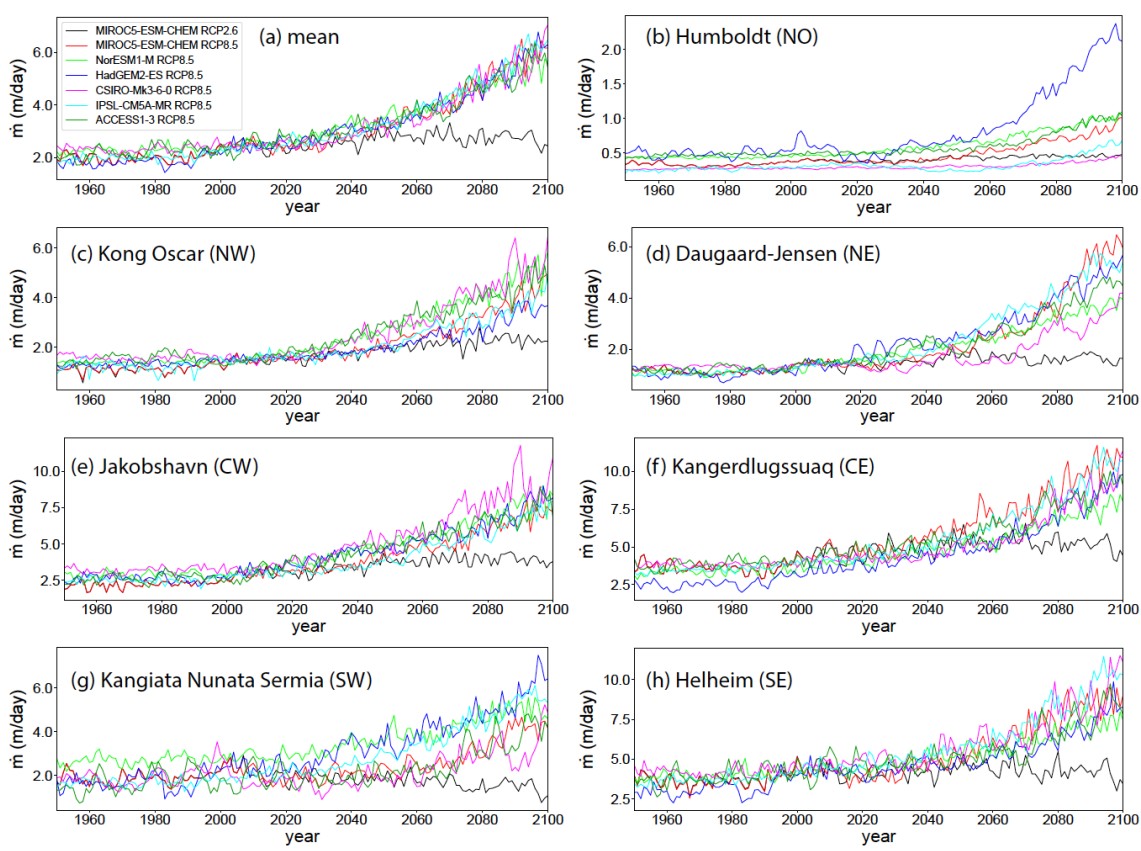

**Figure 10.** Melt rates in the submarine melt implementation. (a) Mean submarine melt rate over the 7 glaciers that are the largest by ice flux in each of the 7 regions, for the CMIP5 models and scenarios listed in Table 1. (b)-(h) Submarine melt rates at the largest glacier by ice flux in each region.

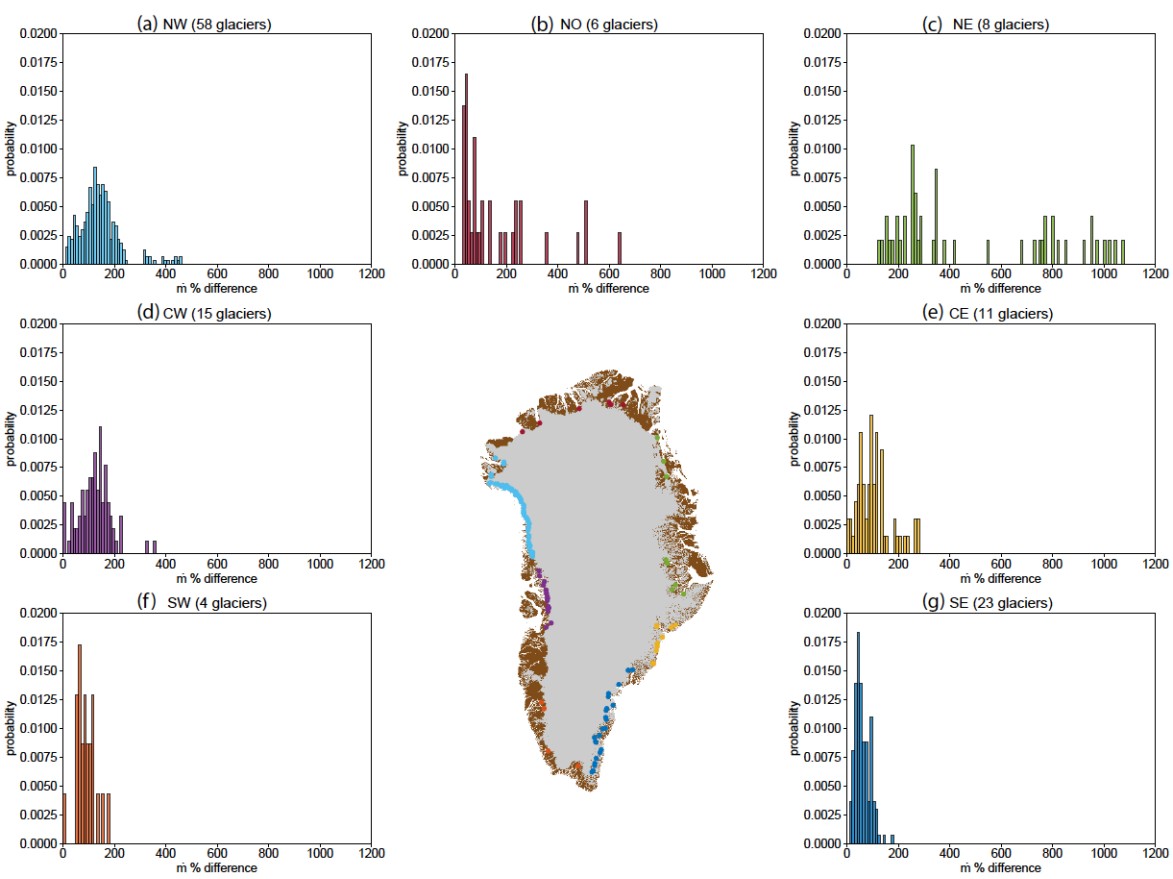

**Figure 11.** Histograms of percentage differences in glacier submarine melt rates between 1995-2014 and 2081-2100 in the 7 ice-ocean sectors for all 6 CMIP5 RCP8.5 scenarios considered.

*Data availability.* The bed topography and bathymetry used in this work may be downloaded from https://nsidc.org/data/IDBMG4 (last access September 2019). Information on the RACMO2.3p2 SMB data can be found at http://www.projects.science.uu.nl/iceclimate/models/greenland.php (last access April 2019). EN4.2.1 oceanographic data is available at https://www.metoffice.gov.uk/hadobs/en4/download.html (last access April 2019). CMIP5 model output is available at https://esgf-node.llnl.gov/projects/esgf-llnl/ (last access April 2019). The MAR based future subglacial discharge projections are available on ftp://ftp.climato.be/fettweis/MARv3.9/ISMIP6/GrIS/ (last access April 2019). TEOS-10 routines may be found at http://www.teos-10.org (last access September 2019). Further information on the ISMIP6 project may be found at http://www.climate-cryosphere.org/activities/targeted/ismip6 (last access April 2019). We intend to make all code used to create these projections freely available on the ISMIP6 GitHub page at https://github.com/ismip (last access September 2019). All of the projection datasets described in this paper are freely available from the ISMIP6 ftp server; access can be obtained by emailing ismip6@gmail.com.

*Author contributions.* DS undertook the majority of the analysis, processing, writing and creation of figures. DF processed past and future runoff and contributed to writing and figures. FS led the ISMIP6 ocean forcing and provided oversight at all stages of the process. HG provided invaluable guidance on the implementation of the described ocean forcing in ice sheet models. CML provided CMIP5 model output and expertise. MM performed the extrapolation of ocean properties into fjords in the submarine melt implementation. XF ran MAR simulations forced by the selected CMIP5 models. SN coordinated the ISMIP6 effort. All authors took part in extensive discussion of the methodology and edited the manuscript.

*Competing interests.* Xavier Fettweis is a member of the editorial board of the journal. Sophie Nowicki is an editor of the ISMIP6 special issue of The Cryosphere.

*Acknowledgements.* We thank Surui Xie, Neil Fraser and an anonymous reviewer for their constructive comments. Donald Slater and Fiamma Straneo were supported by NSF grants 1916566 and 1756272 and by NASA grant NNX17AI03G. Denis Felikson acknowledges financial support from the NASA Postdoctoral Program. Chris Little acknowledges financial support from NSF grant 1513396. Heiko Goelzer has received funding from the programme of the Netherlands Earth System Science Centre (NESSC), financially supported by the Dutch Ministry of Education, Culture and Science (OCW) under grant number 024.002.001. Sophie Nowicki was supported by the NASA Sea Level Change Team and Cryosphere Sciences Programs. Computational resources for performing MAR future projections have been provided by the Consortium des Équipements de Calcul Intensif (CÉCI), funded by the Fonds de la Recherche Scientifique de Belgique (F.R.S.–FNRS) under grant no. 2.5020.11 and the Tier-1 supercomputer (Zenobe) of the Fédération Wallonie Bruxelles infrastructure funded by the Walloon Region under the grant agreement no. 1117545. Thanks to Brice Noël for RACMO2.3p2 output, to Ellyn Enderlin and Michalea King for ice flux datasets and to Jeremie Mouginot for sharing ice sheet basin delineations. All members of the ISMIP6 collaboration are thanked for discussions and feedback, notably at ISMIP6 meetings, with particular thanks to Hélène Seroussi, Alice Barthel and Tim Bartholomaus. We thank the Climate and Cryosphere (CliC) effort, which provided support for ISMIP6 through sponsoring of workshops, hosting the ISMIP6 website and wiki and promoted ISMIP6. We acknowledge the World Climate Research Programme, which, through its Working Group on Coupled Modelling, coordinated and promoted CMIP5 and CMIP6. We thank the climate modeling groups for producing and making

available their model output, the Earth System Grid Federation (ESGF) for archiving the CMIP data and providing access, the University at Buffalo for ISMIP6 data distribution and upload and the multiple funding agencies who support CMIP5, CMIP6 and ESGF. This is ISMIP6 publication number 6.

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
