# Peer review of "21st century ocean forcing of the Greenland Ice Sheet for modeling of sea level contribution"

_The Cryosphere, 2019_

## Referee Comment (RC1) · Surui Xie (Referee) · 23 Oct 2019

Summary: To project Greenland Ice Sheet mass loss due to runoff and ocean property changes in different greenhouse gas emission scenarios, the authors present a modeling effort to apply ocean forcing on a continental scale. Two implementation approaches are presented, including "retreat" and "submarine melt". Both implementations require a parameterization for submarine melting, and the authors consider local ocean velocity and ocean thermal forcing as two primary parameters for submarine melting. The former is implemented through subglacial runoff, and the latter is by ocean temperature. In the retreat implementation, glacier terminus positions are determined by estimated submarine melt rates. It is accessible to all ISMIP6 ice sheet models, but ice dynamics such as glacier advance due to motion or retreat due to calving were not

considered. While the second implementation (submarine melt) takes into account of more factors affecting retreat projections, it is computationally expensive, and some of the considered factors are not currently well understood.

I think this paper is well motivated and well written. It provides a valuable framework for future modeling work of ocean forcing on Greenland Ice Sheet. I only have several fairly minor comments, listed below:

1) Atmospheric-driven runoff and ocean thermal forcing are two primary inputs for the models. While available data or models are limited, some of the assumptions made in this study need to be justified. In section 2.2.3, the runoff bias correction may be necessary to provide a continuous transition from present to future atmospheric forcing, but it may also result in spatial discontinuities, especially when applying a uniform temperature or salinity offset for the entire sector. Figure 2c shows a relatively small bias at Helheim – maybe this is a well monitored glacier so the models perform better? Many other glaciers have much larger values of runoff bias (please see your Figure S1). Could the sector-uniform offset and various bias be major contributors to the difference between different sectors in the projections? This question also applies to the ocean property correction.

2) Ocean temperature is a critical model input in this paper, and is detailed in section 2.3.3. I am curious about the temperature model selection. In Figure 3c, it seems to me that the MIROC5 model produces a quite different temperature profile than the observational EN4 data. Is it rational to use the EN4 data, by simply correcting the bias with a constant offset adding to the entire depth profile? I see that a depth-varying bias correction may lead to unphysical profiles, but is there a reason to choose the mean difference between the specified 200-500 m depth range? According to the authors, this range is perhaps "most relevant to tidewater glacier grounding lines in Greenland". But I feel a slightly different depth range (e.g., 100-400 m) can produce a significantly different offset – especially near the surface. Some discussion on the sensitivity of model to different temperature bias correction may be helpful.
3) For the two implementations, could the retreat history before 2014 be calculated? If this is possible and won't add too much extra work, figures illustrating the historic retreats before 2014 (and maybe comparison with available observations) would improve the integrity of modeling results. Such plots could be added to Figures 4 and 9 as positive retreats.

4) Page 17, line 18: Add "in" after "variability"?

5) Figure 5b: Maybe mark the ~350 m depth point on the dashed red profile? This may help readers understand the effective depth. I had difficulties in understanding the "deepest point" at the beginning – I thought it was rather a shallow (if not shallowest) point at a distance of ~33 km by looking at Figure 5, then I realized that this is a point along the depth of a certain location.

6) Figure 11: Maybe add a vertical line in each panel to mark the largest glacier in the corresponding sector?

Sincerely
* * *

---

## Referee Comment (RC2) · Neil Fraser (Referee) · 31 Oct 2019

The paper investigates the effect of two different parametrisations for ice/ocean interaction, specifically at Greenland's glacier termini, in the context of future ocean/atmospheric conditions as predicted by a range of climate forecast models. This is part of a wider community effort to adequately couple ice sheet models with coupled (ocean/atmosphere) climate models. While both parametrisations consistently predict greatly increased mass loss from the Greenland Ice Sheet under a high greenhouse gas emission forcing regime, the spatial distribution of mass loss varies depending on which climate model is used.

The paper is mostly well written with a clear, direct message and nice figures. I like

the ethos of finding a workable solution to a tough problem at hand and helping the wider community. However, as the authors acknowledge, many aspects of the physical environment are not considered. I felt that this paper really highlighted that major obstacles must yet be overcome before we can expect models to predict future mass loss accurately. I therefore think the results, while certainly valuable, should be interpreted qualitatively rather than quantitatively.

Major comments

1. Thermal forcing, TF, is very simplistic. The authors do well to flag up the shortcomings in section 4.3, but nonetheless there are major shortcomings. The empirical tuning might alleviate this to some extent, but I would still expect the omission of these processes to result in large uncertainties. I understand that sheer necessity offsets these issues to some extent, as the next generation of climate models require parametrisations such as the ones presented here. But I think that any quantitative conclusions about future sea level drawn from those models (which will undoubtedly be very high-impact results) should come with the footnote that ice-ocean parametrisation is still very basic. This is not a criticism of the authors: it's hard to see where major advances will come without much higher resolution AOGCMs.

2. Using annual mean temperature is inappropriate when melt is nonlinear in TF (equation 1). Mean melt is not equal to melt calculated from mean TF. The effect is likely small as the exponent in close to one, but it will result in a systematic error.

3. My understand is that, if dL = melting + calving, retreat represents both terms while submarine melting represents only the first term. This should be made more explicit earlier on. Some of the language makes it a bit unclear what the inputs and outputs are for each parametrisations, and can seem at odds with Equations 1 and 2. I comment below on the specific instances of this.

4. Using EN4 for bias correction makes sense in theory, but do you have a sense of how many direct observations actually influence the EN4 gridded product for the

regions/times of interest? EN4 has had issues in the Labrador Sea, and the EN4 temperature profile (Fig3c) is not a good representation of typical SE Greenland stratification (there should be a subsurface temperature maximum). You could add a figure in the supplementary material showing the mean EN4 confidence weightings for each ocean sector. Could bias correction be done instead using the available CTD profiles from each sector?

Minor comments

Some of these are stylistic comments which the authors are entitled to disagree with.

P1L8: It's misleading to say that retreat is a function of submarine melting, do you mean subglacial runoff? I read this to mean that one parametrisation feeds into the other. This is related to major comment 3.

P1L9: You should give RCP2.6 and 8.5 formal definitions, if not here then in the introduction or methods.

P2L21: Can you be more quantitative about the number of ice shelves than "a handful"?

P2L25: Perhaps also worth mentioning here that since these regions are very poorly observed, especially in winter, large uncertainties remain with regards to Greenland fjord/shelf processes (i.e. while you correctly state that these processes are not captured in models, we still don't know exactly what we are trying to capture!).

P3L13: I would considering moving this first paragraph to the introduction. I see that it leads nicely into the second paragraph in 2.1, nonetheless when I finished reading the nice introduction it was frustrating to find myself reading what was essentially just more introduction.

P4L1: If submarine melt rate is denoted by m^dot and dL is linear in submarine melt, then should this not be make explicit in the expression for dL? Otherwise, perhaps more careful language should be used. Again this ties into major comment 3.

[Figure]

P4L14: Personally I don't like multiplication signs in formulae, and I think it would read better if you dropped them.

P4L16: Refer to section 2.3.1 instead of "further below".

P4L22: I'd change "even in future projections" to "particularly in future projections" since one would anticipate annual and summer means to converge as summer becomes longer.

P5L25: I really like this section on bias correction. Very clearly thought out and explained. It might be worth citing Menary et al. 2015 GRL, who explore CMIP5 temperature and salinity biases in the Labrador Sea west of Greenland, to underline your motivation.

P10L5: I understand that certain simplifications are necessary for these parametrisations to work in coarse climate models, but this paragraph completely ignores a lot of the research into fjord/shelf hydrodynamics. Much of these shortcomings are acknowledged later in section 4.3, but I think they should be made clear up front.

P10L22: If TF used in equation 1 differs from the TF used in equation 2 then perhaps they should be given different symbols or subscripts.

P10L31: See major point 2.

P12L19: I'd remove the word "however" as it isn't necessary.

P13L18: This appears to be a strong argument for using more than one RCP2.6 model in your experiment.

P15L23: This seems to imply the thermal forcing is an input for the submarine melt regime only, when in fact it is an input for both. These two sentences could be rewritten to make it absolutely clear what the input and output variables are for each regime.

P15L25: Change "...as they see fit" to "... as required" or similar, to avoid referring to a model as "they".

P15L25: Sentence starting "Each implementation...": This sentence is really jarring and frankly bizarre. If it wasn't interesting you wouldn't be writing a paper on it!

P16L31: Typo, "large uncertainty in..."

P18L3: Even without dense overflows, the properties of the water trapped behind the sill can (and will) be modified by downward mixing of buoyancy from the upper layers.

P18L14: The paragraph could also mention wind-driven heat delivery via the internal wave field, which has been found to deliver ocean heat to fjords in Greenland. Also, ideally the submarine melt parametrisation would capture (horizontal) ocean current speed adjacent to glacier termini, which we know is related to e.g. fjord width (i.e. Jackson et al. 2018) and impacts melting.

Fig3c: EN4 temperature profile looks suspect, what are the EN4 confidence weightings here? (major point 4)

Fig6a: Is this figure saying that in 2100, ocean water will have flooded beneath the interior if the ice sheet? If so, this is a major result which should be flagged up in the text.

Fig9: To me negative retreat implies advance, so I'd change either the axis label (to "frontal position"?) or sign. Figure 10 uses positive values to denote mass loss, it'd be better if they were consistent.

Fig10f: There's a missing dot above the m labelling the y-axis.

Overall, an important step towards the goal of coupled air-sea-ice climate models (but there is still a way to go).

Kind Regards,

Neil Fraser

Please also note the supplement to this comment:
https://www.the-cryosphere-discuss.net/tc-2019-222/tc-2019-222-RC2-supplement.pdf
* * *

---

## Referee Comment (RC3) · Anonymous Referee #3 · 4 Nov 2019

General Comments: The manuscript details the implementation and results of applying two different ocean forcing strategies (termed the retreat and submarine melt implementations) to a suite of AOGCMs contained in CMIP5 for use in the upcoming ISMIP6 to inform the next IPCC report (AR6). The authors present the model parameterizations, including their motivations and the limitations of each implementation. Then, they apply the implementations to a set of CMIP5 models and present the resulting forcing parameters (subglacial runoff and ocean thermal forcing) and model projected retreat and submarine melt rates.

The work presented in the manuscript presents an important step forward for modeling ice sheet response to various warming scenarios and consequent contributions to sea level rise. The authors do a great job presenting their model implementations and the

motivations for each, and overall the ideas are well-organized and logically presented. However, as currently written the manuscript suffers from several key problems, outlined below, that must be addressed before it is suitable for publication.

1. Lack of clarity. Specifically, a clearer introduction would eliminate a lot of potential confusion later in the manuscript surrounding how ocean boundary conditions influence ice sheet mass changes and how these are both important in models. As currently written, the distinction between process understanding within the field and effective modeling of those processes is unclear. There is a constant switching between observations and modeling that leaves the reader guessing which one is currently being discussed, and descriptions of how the processes are linked would greatly improve clarity for the modeling components of the writing. In some cases, significant detailed disciplinary knowledge is required to explain and justify the limitations and assumptions used within the model.

2. The discussion section is underdeveloped. Many interesting discussion points are presented and then left hanging without further exploration. Similarly, some of the assumptions and simplifications presented throughout the methods and results are not fully discussed, even where more information is available to inform a discussion (e.g. the magnitude of bias corrections and their interpretation; the influence of uncertainty in bathymetry; the potential uncertainty stemming from the assumption that submerged ice area remains constant).

3. The writing needs work both for content and grammar. As previously noted, the writing is overall unclear, with a lot of extraneous words at the expense of sufficient content in some places (e.g. p6 line 25: what is "inefficient" about current parameterizations? Are they computationally expensive or are they simply ill constrained?). Passive voice, dangling modifiers, and phrases not clearly linked to their parent idea are prevalent throughout, along with unneeded words and phrases (including connecting phrases such as thus, therefore, however, then, here). These detract and distract from the real power of the manuscript. Comma usage is quite poor (it is pointed out completely

only in the abstract line comments, below), including inconsistent use of the Oxford comma and missing and incorrect comma usage, particularly around non-compound sentences. Lastly, please proofread for subject-verb agreement, overall grammar, and consistency in formatting (e.g. capitalization of the Greenland Ice Sheet). This includes literature references, which are currently not consistently cited for the same ideas and lack cohesive formatting throughout (i.e. in-text citations are in random orders) and section references, which are an interesting mix of high level and lower-level references and do not consistently point to the most logical section (thus leading to confusion rather than clarity).

Specific ("line") comments:

Abstract:

p1 Line 1: Please expand on what oceanic "changes" you mean p1 Line 3: Provide some examples of what you mean by "key physics" and make "limitations in processing understanding" less ambiguous p1 Lines 3 and 15: unnecessary comma p1 Line 9: comma before respectively

Introduction:

p1 Line 20: passive voice p2 Line 2: dangling modifier p2 Line 6: comma needed after thus p2 Line 8: CMIP6 is used as an acronym before it is defined p2 line 19: it would be helpful for the reader to succinctly describe what CMIP is in this paragraph, as you have done for ISMIP. p2 Lines 21-22: ice shelves and floating ice tongues (and remove comma – not a compound sentence) p2 Line 22: clarify model design, or it sounds like you are designing the ocean forcing itself p2 Line 32: inconsistent reference ordering p3 first full paragraph: clean up language and extra words p3 Line 9: use of Oxford comma needs to be removed or added throughout manuscript

Methods:

Overview:

[Figure]

p3 first paragraph: many of these ideas are repetitive with information presented in the introduction (though with different sets of references). The temporal words (past decade, since) are misleading relative to the information presented (warming in the late 1990s) and references (2010). p3 line 27: the links between calving rate, glacier retreat, and ice sheet mass loss have only been tenuously drawn. Please include a clearer description of these physical processes prior to discussing their modeling. p3 line 31-32: both italics and quotations does not match the abstract formatting p3 Line 33: taking part in ISMIP? p4 eq 2: for the reader not intimately familiar with Slater et al 2019, another sentence about kappa (how it is calibrated, under what conditions it is applicable/scalable) would be helpful. p4 eq 1 and 2: switching the order of presentation of these two equations would provide order consistency with the presentation of the retreat, then submarine melt, implementations throughout the text p4 line 34: the "or CMIP6" is confusing here. It might be helpful to instead note above, where you are addressing your use of CMIP5 inputs, that the process would be identical for using CMIP6 inputs.

Atmosphere:

p5 line 3: define acronym MAR p5 line 4: the use of "physically downscaling" is confusing, especially given the later statement that the downscaling is done statistically. Removing "physically" would improve clarity. p5 line 8: repetitive statement p5 section 2.2.2: If I am understanding correctly, hydrologic drainage basins are determined based on hydrologic potential (fine). Then, subglacial runoff is determined using surface runoff for those previously delineated basins. I think the authors need to better support and acknowledge the inherent assumptions here, including: 100% of surficial runoff reaches the bed and the surficial runoff reaches the bed with a similar spatial distribution, such that subglacial drainage basins with surface melt volume are appropriate for estimating subglacial melt volume. I would also like to see the use of f=1 substantiated. p6 line 5: how is $Q_j(1995-2014)$ for RACMO or PROJ calculated? Is it a mean? Median? Cumulative? p6 line 9: it would be helpful to provide some basic

information on the bias corrections within the text (e.g. range and median+uncertainty). p6 lines 9-10: "we note that it might be thought preferable" is very wordy and passive language p6 line 11-13: this sentence could also be made stronger, particularly by quantifying the insignificance of the difference between RACMO and MAR (and noting the range of annual variability).

Ocean:

p6 line 16: needs parenthesis p6 line 22: what atmospheric process? For calculating surface runoff? p6 line 30-31: remove "details in" p7 line 10: quantify "some distance". What criteria did you use to determine the extent of the sector beyond the shelf break? p7 line 13: quantify "coarse resolution" p8 line 7: as for the runoff, it would be helpful to present the range and some statistics (range and mean/median with uncertainty) on the applied bias corrections

Retreat Implementation:

p8 line 14: in the last equality, how are the units converted from salinity to temperature? p8 line 16: how were these constants determined, and are they valid for use here? p8 line 25: the thermal forcing itself is actually described in section 2.4.1, not section 2.3 p8 line 31: I'm not sure why this equation is presented independently of equation 2, which is the general form. I don't think this equation is substantially different enough to warrant a second presentation, particularly since the text notes the projection is relative to 2014. p9 lines 1-5: the information on kappa presented here should be included the first time the equation is mentioned.

Submarine melt implementation:

p10 line 10: what criteria were used (e.g. slope) to determine when a feature was large enough to be considered "blocking"? p10 line 29: no capitalization on where p11 line 2: I'm not entirely convinced of the validity of using JUST the ocean bottom value for the thermal forcing, particularly if it's not the highest thermal forcing within the vertical

profile. What rationale can be provided to suggest this won't underestimate melt rates? p11 line 14: typo

Results:

p12 line 6: section heading misses emphasis within section on glacier runoff p12 line 7: the use of "and" is confusing and suggests multiple runoff values are prescribed. Perhaps "each tidewater glacier/hydrological drainage basin" or "each tidewater glacier using its hydrological drainage basin" p12 lines 15-18: these sentences are more speculations than observations p12: The switch in referencing between largest glacier by flux and region between the text and figures is confusing. Suggestions to increase clarity are: add the glacier names to Figure 3 when the sectors are introduced and more importantly to Figure 7 (a and b) where the data shown is actually for individual glaciers and not the entire region. p12 line 30: inappropriate semicolon p13 line 3: section 2.4.1 refers to the section on thermal forcing, making this statement confusing. I've stopped noting odd section references after this point p13 line 16: the supplementary panel figures are not labeled with letters p14 lines 23-24: inconsistent use of sector names (e.g. At Humboldt Glacier (NO), little increase...) p15 line 3: The total count note is helpful, but confusing if you don't know offhand that 58 is the number of glaciers.

Discussion:

p15 line 25: anthropomorphizing of ice sheet models ("they see fit") p15 line 26: this statement implies that you have not already contrasted modeled ice sheet response between the two implementations p16 line 1: this suggests that averaging retreat over a population of glaciers resolves the fact that we cannot currently accurately represent calving or fully account for bathymetry in models of glacier termini. I would strongly disagree. Regional averaging may improve our modeled representation of retreat, which is of scientific import for further modeling and informing future investigations, but it does not fundamentally "ameliorate these issues". p16 lines 12-21: this reads as results, not discussion p16 line 23: subglacial runoff and ocean thermal forcing cannot be compared directly, as they are volume and temperature measures, respectively. p16 line 25-26: impact on what? retreat? mass loss? p16-17 lines 27-2: I would like to see this idea explored further (and clarification on why the authors switch from generalized "retreat and submarine melt projection" statements to just the "retreat implementation"). What are the implications of this compensation across model suite versus across retreat scenario comparisons? p17 lines 3-8: again, develop this idea further p17 lines 18-19: rewrite – currently not a sentence p17-18 lines 33-6: This paragraph leaves out the problem of incomplete bathymetry observations, a key area where improved models will still be limited by lack of observations. p18 line 10: other processes DO play a role, not MAY play a role. Perhaps the authors mean to emphasize that the other processes MAY play a SUBSTANTIAL role? p18 line 13: dash needed between ice sheet and ocean p18 lines 17-20: can you make the argument that the physics of plumes are well understood if we severely lack constraints for key constants?

Figures:

Overall: It would be helpful to use a different color scheme for showing comparisons of model runs than those colors used for plotting different sectors. This would allow the reader to more readily distinguish between sector-based results versus those averaged over the entire ice sheet that show model variability.

Figure 2: a zoom-in of the shaded portion of panel c (which is unlabeled but presumably indicates the time period used for the bias correction) is needed. As shown, it is difficult to see the similarities and differences between the datasets used to make the bias correction, and the zoomed out version suggests some apparently large differences between RACMO and MAR that are not substantially addressed within the text.

Figure 3: Add the resolution of the climate model shown in panel a for clearer comparison with panel b (which has a stated resolution).

Figure 5: a is missing units; the yellow and red points/lines are not labeled

Figure 6: b-why is such a large portion of the ice sheet showing no-data (entire drainage basins do not have subglacial runoff values)?

Figure 7: caption – subject verb agreement; b – is this also showing the largest glacier by ice flux for that sector? Also, the figure is missing units

Figure 10: The labeling with glacier name and region is quite helpful

Figure S1: The SE and NW colors are difficult to tell apart. Figures S3-S8: why not utilize some of the white space where there are no subplots as adequate space for the legend (particularly where it has been separated across multiple plots)

Acknowledgements: Michalea's name is spelled wrong

---

## Author Comment (AC1) · 13 Jan 2020

We thank the reviewers for their thoughtful and constructive reviews and are pleased that the manuscript was well received. Please find attached our response to reviews, revised manuscript and supporting information, and a version of the manuscript with changes marked.

Please also note the supplement to this comment:
https://www.the-cryosphere-discuss.net/tc-2019-222/tc-2019-222-AC1-supplement.zip

---

## Author Response (AR1)

**Response to editor for tc-2019-222: "21st century ocean forcing of the Greenland Ice Sheet for modeling of sea level contribution"**

*Thank you very much for the prompt response on our revised manuscript and for taking the time to serve as editor for our submission. Below we detail how we have made all of the minor changes suggested. Our responses are in **blue italics**. In the second introductory paragraph we have now also added references to the recently submitted papers describing in the ISMIP6 protocol (Nowicki et al., 2020a) and the Greenland sea level projections (Goelzer et al., 2020).*

page 1, line 8: I thought having the first use of 'retreat' and 'submarine melt' as the names of the parameterisations in quotes made a helpful distinction, actually.
*Changed as suggested.*

p2,l8: use of 'our' implying ownership of ISMIP6 could be misleading. "the" would be simple
*Agreed – changed as suggested.*

p2,l16: the only GrIS ocean forcing you're supplying comes from CMIP5 models, I think this sentence needs to be shaded - or just removed entirely? The Summary at the end had some similar sentences that I thought were clearer
*In the end we have supplied some CMIP6-derived GrIS ocean forcing to the ice sheet modelers, but these are more fully described in separate papers (Nowicki et al., 2020a,b) and so it is probably cleanest to remove this sentence as suggested. The use of CMIP5 vs CMIP6 models is also discussed in P5L13-P5L18.*

p4,l1: "resolution of technical capability" should be 'or', I guess?
*Yes – thank you for spotting this.*

p4,l5: 'capabilities', plural, reads better to me
*Changed as suggested.*

p6,l22: "may deviate": no "may" about it, they certainly do, for many quantities
*Agreed, we have revised this sentence accordingly.*

p7,l17: "methodology" would be better as "practice". There /are/ differences in the methodology, but they have negligible impact in practice.
*Yes – thanks for this suggestion – now changed.*

p9,l21: "In-keeping" is normally two separate words, not hyphenated
*Now fixed.*

p12,l7: "Lastly": this is a not really a logical continuation of the list of things, I'd just say "We note that". You've already had a "Lastly" in this paragraph too (p12,l2)
*Changed as suggested.*

p18,l28: It's not just that CMIP5 AOGCMs don't resolve the fjords (leaving the reader to infer that maybe CMIP6 ones do) - you used "CMIP" as a general term before, I'd use it again here to make it clear that this isn't a problem with one specific CMIP iteration.
*Yes – thanks for the suggestion.*

Literature cited

Goelzer et al., 2020: The future sea level contribution of the Greenland ice sheet: a multi-model ensemble study of ISMIP6, *The Cryosphere Discussions*, 1-43, doi: 10.5194/tc-2019-319

Nowicki et al., 2020a: Experimental protocol for sea level projections from ISMIP6 standalone ice sheet models, *The Cryosphere Discussions,* 1-40, doi: 10.5194/tc-2019-322

Nowicki et al., 2020b: Contrasting contributions to future sea level under CMIP5 and CMIP6 scenarios from the Greenland and Antarctic ice sheets, submitted to *Geophysical Research Letters*

[revised manuscript text omitted]